# Solute exchange through gap junctions lessens the adverse effects of inactivating mutations in metabolite-handling genes

Stefania Monterisi[1]*, Johanna Michl[1], Alzbeta Hulikova[1], Jana Koth[2],
Esther M Bridges[2], Amaryllis E Hill[1], Gulnar Abdullayeva[2], Walter F Bodmer[2],
Pawel Swietach[1]*

[1]Department of Physiology, Anatomy & Genetics, University of Oxford, Oxford,
United Kingdom; [2]MRC Weatherall Institute for Molecular Medicine, John Radcliffe
Hospital, Oxford, United Kingdom

*For correspondence:
stefymonte@hotmail.com (SM);
pawel.swietach@dpag.ox.ac.
uk (PS)

Reviewing Editor: Jean X Jiang,
The University of Texas Health
Science Center at San Antonio,
United States

**Abstract** Growth of cancer cells in vitro can be attenuated by genetically inactivating selected metabolic pathways. However, loss-of-function mutations in metabolic pathways are not negatively selected in human cancers, indicating that these genes are not essential in vivo. We hypothesize that spontaneous mutations in 'metabolic genes' will not necessarily produce functional defects because mutation-bearing cells may be rescued by metabolite exchange with neighboring wild-type cells via gap junctions. Using fluorescent substances to probe intercellular diffusion, we show that colorectal cancer (CRC) cells are coupled by gap junctions assembled from connexins, particularly Cx26. Cells with genetically inactivated components of pH regulation (*SLC9A1*), glycolysis (*ALDOA*), or mito-chondrial respiration (*NDUFS1*) could be rescued through access to functional proteins in co-cultured wild-type cells. The effect of diffusive coupling was also observed in co-culture xenografts. Rescue was largely dependent on solute exchange via Cx26 channels, a uniformly and constitutively expressed isoform in CRCs. Due to diffusive coupling, the emergent phenotype is less heterogenous than its genotype, and thus an individual cell should not be considered as the unit under selection, at least for metabolite-handling processes. Our findings can explain why certain loss-of-function mutations in genes ascribed as 'essential' do not influence the growth of human cancers.

## Editor's evaluation

This innovative study is of potential interest to a broad readership of cancer biology by addressing an important mechanism regarding how spontaneous mutations in cancer cells affecting meta-bolic pathways do not necessarily result in a functional defect thanks to gap junctionally-mediated exchange of metabolites.

## Introduction

The role of somatic evolution in advancing cancers is well-established (**Nowell, 1976**; **Merlo et al., 2006**) and can be modeled mathematically (**Wölfl et al., 2022**; **Johnston et al., 2007**). Its central paradigm asserts that mutations benefiting cancer cells undergo positive selection and appear enriched in cancers. Cells carrying such 'advantageous' genetic changes may evade the normal checks and controls that restrict proliferation (**Tomlinson and Bodmer, 1995**), or become resistant to the harsh microenvironments present in tumors (**Gillies et al., 2012**). Conversely, mutations that inactivate critically important processes are predicted to emerge less frequently than the random mutation rate (**Greenman et al., 2006**). A number of in vitro knockout (KO) screens have identified

genes deemed essential for cell survival (*Wang et al., 2015*; *Blomen et al., 2015*), among which are genes involved in metabolic pathways (*Sinkala et al., 2019*; *Denko, 2008*). Loss-of-function mutations in these processes are predicted to undergo negative selection in human cancers (*Zapata et al., 2018*; *Bailey et al., 2018*), but this phenomenon is exceedingly rare in vivo, equating to only tens of genes (*Martincorena et al., 2017*) such as those involved in protein synthesis and immune-exposed epitopes, but not metabolite handling (*Zapata et al., 2018*). The reason for the discrepancy between the interpretation of in vitro screens and in vivo observations remains unclear, but it highlights fundamental differences in the notion of gene essentiality under culture conditions and in human tumors. We hypothesize that inactivating mutations in key metabolic pathways may be rescued by access to functional proteins in neighboring wild-type (WT) cells. This scenario is more likely to happen in tumors bearing spontaneous mutations, as compared to mutagenized mono-cultures in vitro. Thanks to this rescue effect, the aforementioned mutations will not incur a functional deficit to the mutation-bearing cell, which - in turn - will not undergo negative selection. A plausible means of rescue could involve connexin-assembled gap junctions that allow metabolite exchange between cells across the continuous cytoplasmic space or syncytium (*Dovmark et al., 2018*; *Dovmark et al., 2017*; *Aasen et al., 2016*; *Nicholson, 2003*). Indeed, early work on gap junctions in fibroblasts provided evidence for metabolic cooperation, whereby connected cells can compensate for complementary deficiencies, such as in enzymic activity (*Gilula et al., 1972*; *Pitts, 1998*; *Fujimoto et al., 1971*). However, the prevailing paradigm at the time stated that cancer cells do not form gap junctions (*Loewenstein and Kanno, 1966*), and thus cannot benefit from metabolic cooperation. The prominence of gap junctions in cancer has since been revised (*Dovmark et al., 2018*; *Dovmark et al., 2017*; *Aasen et al., 2016*; *Nicholson, 2003*), but the feasibility of metabolic rescue remains to be tested. In the proposed model, an inactivating mutation arising spontaneously within a coupled network of cancer cells may not emerge as deleterious when the affected cell can access metabolite-handling enzymes or transporters in neighboring cells across connexin-assembled channels. If, however, the same mutation is introduced experimentally into all cells (as is typical with in vitro assays), diffusive exchange is unable to restore metabolic function, irrespective of coupling strength. We speculate that the diffusive exchange of solutes via gap junctions can explain why some metabolic processes may be essential for survival in vitro, but do *not* undergo negative selection in human cancers. This model shares similarity to the bystander effect, whereby toxins or therapeutic agents can spread across a tumor via gap junctions (*Aasen et al., 2016*; *Spray et al., 2013*; *Mesnil et al., 1996*; *Pitts, 1994*; *Bi et al., 1993*). Metabolic rescue of gene mutations would add another dimension to the role of gap junctions in cancer by addressing their longer-term consequences on carcinogenesis, and offering an explanation for the scarcity of negatively selected genes.

This study used colorectal cancer (CRC) cells to test our hypothesis that diffusive exchange can rescue cells carrying inactivating mutations in apparently critical genes. We first assessed the degree of cell-to-cell coupling in a panel of CRC cells and identified the major connexin isoforms responsible for assembling these conduits. We then genetically inactivated specific metabolic processes that play important roles in cancer to test whether coupling onto WT cells can compensate the genetic deficit. The inactivated functions included (i) Na$^+$/H$^+$ exchanger-1 (*SLC9A1*), a pH regulator at the plasma membrane that is critical for survival under acidic conditions (*Counillon et al., 2016*); (ii) aldolase A (*ALDOA*), a glycolytic enzyme that is critical for the Warburg effect (*Ritterson Lew and Tolan, 2012*); and (iii) NADH:ubiquinone oxidoreductase core subunit S1 (*NDUFS1*), part of complex I required for oxidative phosphorylation (*Urra et al., 2017*). In support of our hypothesis, we found that co-culturing genetically-altered cells with WT counterparts rescues the functional deficit by allowing access to operational proteins in neighboring cells. Moreover, we confirm this rescue effect in vivo using co-culture xenograft models. Based on our findings, we propose that connexin-dependent metabolite exchange lessens the deleterious effects of mutations in genes related to metabolism.

## Results
### Screening colorectal cancer cells for connexin expression
Microarray data from 79 CRC lines (*Wilding et al., 2010*) were analyzed for the expression of connexin (Cx) genes. Expression in a subset of CRC cells was detected for *GJA1* (coding for Cx43), *GJA3* (Cx46), *GJB1* (Cx32), *GJB2* (Cx26), *GJB3* (Cx31), *GJB5* (Cx31.1), and *GJC1* (Cx45) (*Figure 1A*).

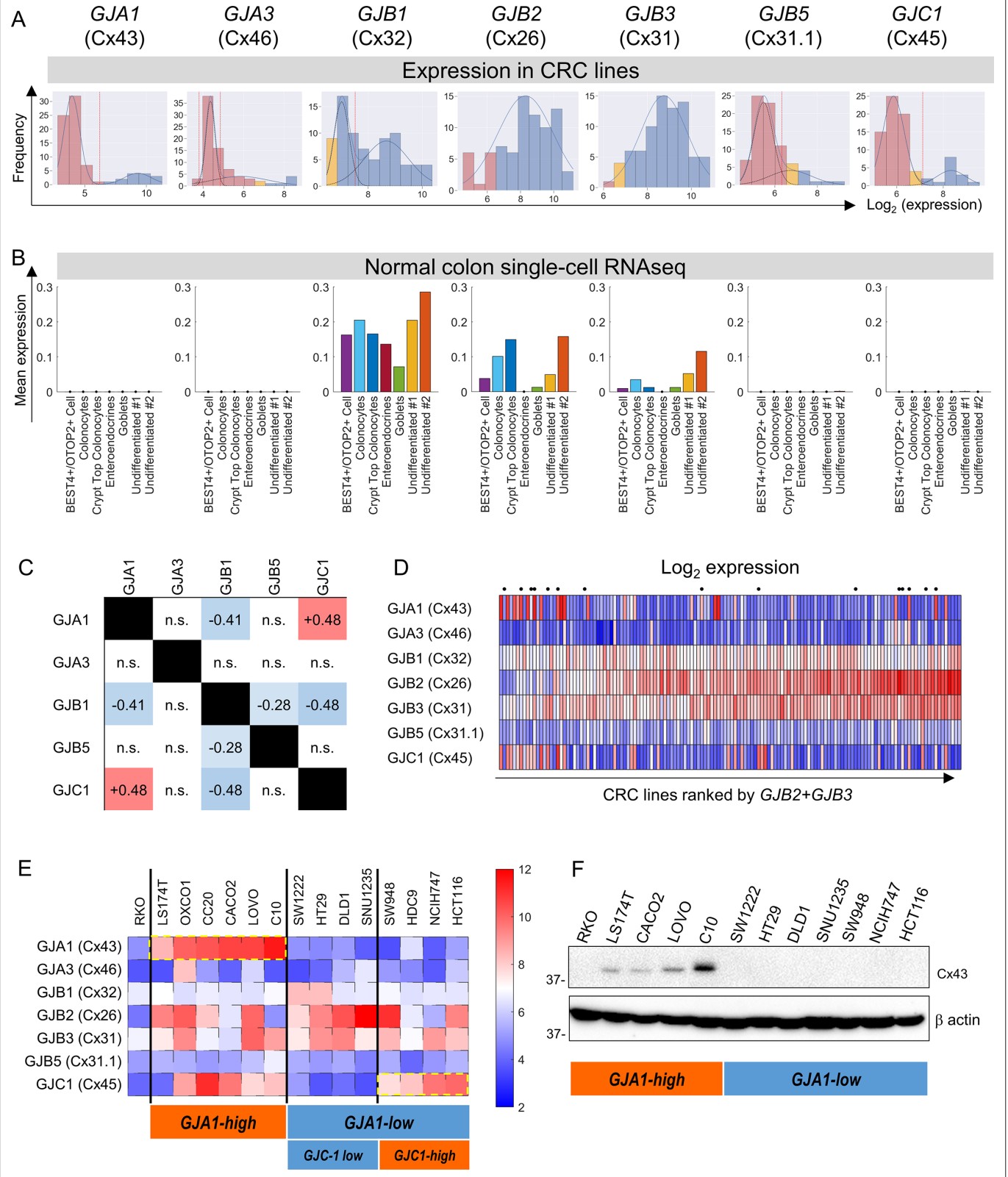

**Figure 1.** Connexin isoform expression in colorectal cancer (CRC) cells. (**A**) Microarray data from 79 CRC cell lines analyzed for message level of seven connexin genes. Frequency distributions for log₂-transformed data. A Gaussian mixture modeling (GMM)-based analysis (GMMchi) is used to determine whether the distributions are bimodal or unimodal. Vertical red line is the cutoff threshold separating low/high groups. Pink bars refer to background levels, and yellow bars refer to near-background levels, based on a separate analysis of the overall pattern of gene expression observed

*Figure 1 continued*

in the cell lines. The cutoff thresholds for the difference between low and high expression are $2^{6.2}$, $2^{4.9}$, $2^{7.5}$, $2^{6.3}$, and $2^{7.1}$ for *GJA1*, *GJA3*, *GJB1*, *GJB5*, and *GJC1*, respectively. (**B**) Analysis of single-cell RNAseq datasets for normal colon obtained from the GSE116222 dataset available at the Gene Expression Omnibus. Bars show mean expression levels by cell type. (**C**) Two-by-two table shows correlation between bimodally distributed connexin genes. Numbers refer to correlation coefficient for significant ($p<0.05$) gene pairs (Fisher's exact test). (**D**) Log$_2$-transformed expression data ranked by the sum of *GJB2* and *GJB3* expression. Cells selected for further studies are indicated by a dot above the heatmap. (**E**) Heatmap replotted for the selected 15 cell lines, grouped by *GJA1* and *GJC1* expression, relative to threshold determined from GMM analysis. (**F**) Western blot for Cx43, showing agreement between protein levels and gene expression profiles.

The online version of this article includes the following source data and figure supplement(s) for figure 1:

**Source data 1.** Full-length scans of blots for *Figure 1F*.

**Figure supplement 1.** Application of the GMMchi pipeline in purifying non-tumor expression from bulk tumor expression in the TCGA patient samples.

**Figure supplement 2.** Principal component analysis (PCA) of microarray data for connexin gene expression, covering 79 colorectal cancer (CRC) lines.

Gaussian mixture modeling (GMM) determined that the distributions of *GJA1*, *GJA3*, *GJB1*, *GJB5*, and *GJC1* message among CRC lines were bimodal, such that cell lines could be grouped as high- or low-expressing. In contrast, messages for *GJB2* and *GJB3* were unimodally distributed among the cell lines. Analysis of TCGA datasets confirmed bimodality for *GJA3*, *GJB5*, and *GJC1* expression (*Figure 1—figure supplement 1*). Single-cell transcriptomics (*Fawkner-Corbett et al., 2021*) indicated expression of *GJB1*, *GJB2*, and *GJB3* in normal colonic epithelial cells (*Figure 1B*). *GJB2* and *GJB3* had previously been identified in normal colorectal epithelium (*Sirnes et al., 2015*; *Dubina et al., 2002*). Overall, *GJB2* and *GJB3* are constitutively expressed in CRC cell lines, whereas *GJA1*, *GJA3*, or *GJC3* were found in a subset of lines, presumably due to mutations or stable methylation differences in their promoter regions. Two-by-two table analysis for bimodally distributed connexin genes indicated positive correlations between *GJA1* and *GJC1*, and negative correlations for pairs *GJB1-GJA1*, *GJB1-GJB5*, and *GJB1-GJC1* (*Figure 1C*; see also principal component analysis in *Figure 1—figure supplement 2*).

*Figure 1D* shows the expression of connexin genes, ranked by combined *GJB2* and *GJB3* message. Fourteen lines were selected for further functional measurements (*Figure 1E*). Since *GJA1* expression accounted for considerable variation, cell lines were categorized as *GJA1*-high and *GJA1*-low. The *GJA1*-low group could be subdivided further as *GJC1*-low and *GJC1*-high. This classification was confirmed at protein level using antibodies against Cx43 (*GJA1*) (*Figure 1F*). RKO cells were chosen as a negative control because of negligible expression of major Cx genes.

## Cx26 (*GJB2*) channels are a major route for diffusive exchange between CRC cells

Solute diffusion between cells was inferred from fluorescence recovery after photobleaching (FRAP) of fluorescent calcein (*Swietach and Monterisi, 2019*). A cell in the middle of a coupled cluster was bleached to ~50% of resting signal, and the subsequent recovery of fluorescence (due to dye ingress from neighboring cells) indicated the degree of coupling. Fluorescence recovery was detected in *GJB2*-expressing SNU1235 but not in connexin-negative RKO cells (*Figure 2A*). FRAP recordings were quantified in terms of an apparent permeability constant for calcein (*Figure 2B*), and the role of Cxs was confirmed from the inhibitory effect of carbenoxolone (100 µM), a broad-spectrum Cx blocker (*Figure 2C*). The acquired images supply data on cell geometry and time courses of calcein fluorescence for estimating permeability to calcein, albeit with assumptions detailed in Appendix 1. Cell-to-cell calcein permeability correlated best with *GJB2* expression (*Supplementary file 1*). Moreover, *GJB2* knockdown ablated functional coupling, whereas siRNA against *GJB3* did not (*Figure 2D*; see *Figure 2—figure supplement 1* for knockdown confirmation). In *GJA1*-high Caco2 cells, functional coupling was ablated with siRNA against *GJA1* (*Figure 2E*; see *Figure 2—figure supplement 2* for knockdown confirmation). Note that there was a trend for *GJB3* knockdown to increase overall connectivity in DLD1 and SNU1235 cells, which may represent compensatory changes in the expression of other connexins or closeness of cell–cell contacts. Western blot analysis indicated some degree of interaction between the expression of particular pairs of connexins, but the effects were, at best, modest (*Figure 2—figure supplement 1*).

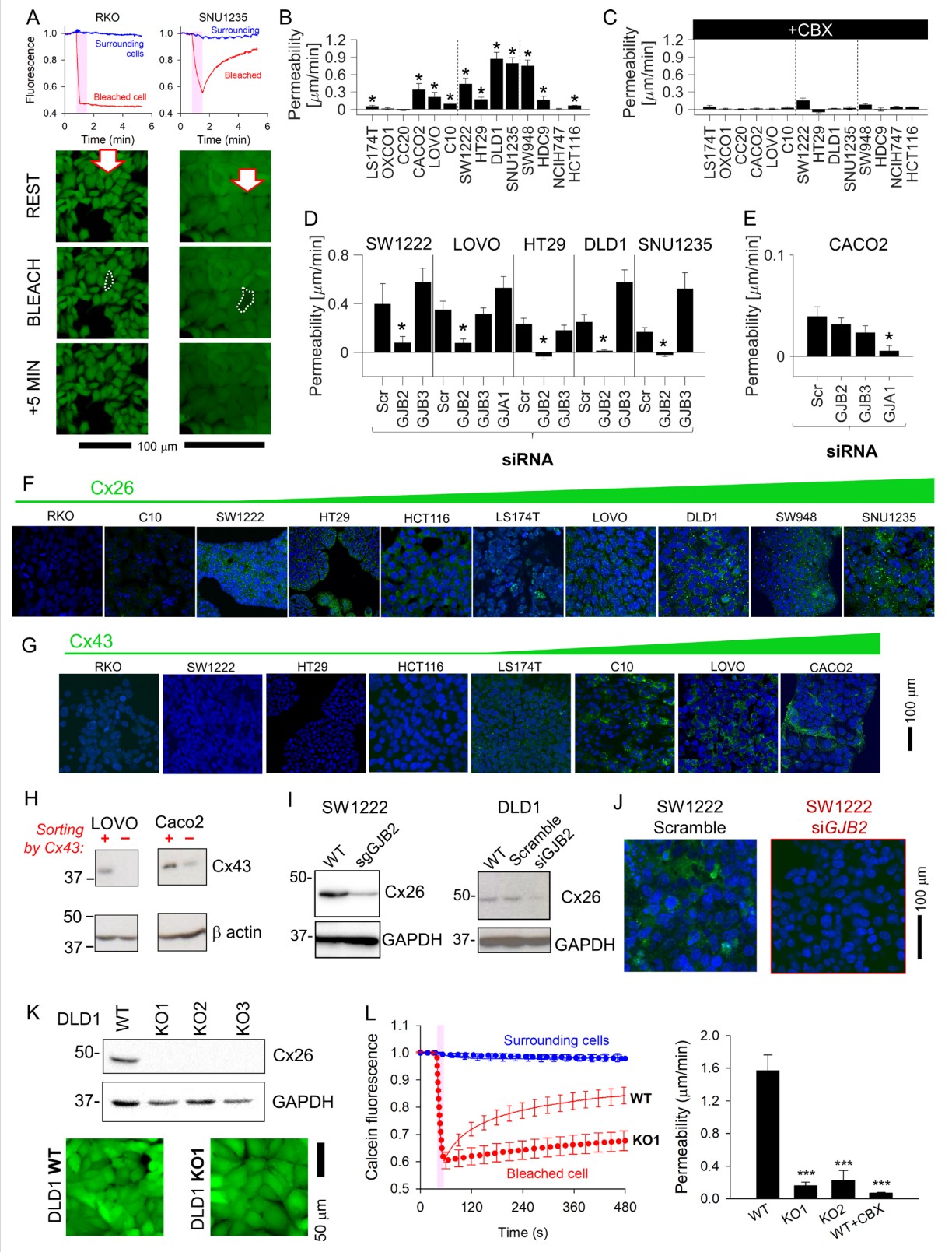

**Figure 2.** Connexin isoforms underpinning cell-to-cell coupling in colorectal cancer (CRC) cells. (**A**) Fluorescence recovery after photobleaching (FRAP) protocol for interrogating the apparent cell-to-cell permeability to calcein in RKO cells (connexin-null) and SNU1235 cells (Cx26-positive). Images taken before bleaching (resting), immediately after bleach, and 5 min after bleach. (**B**) Apparent permeability to calcein (mean ± SEM) in CRC monolayers; * denotes significant coupling (*t*-test). For each experiment, measurements were obtained from at least five independently grown monolayers, with

*Figure 2 continued on next page*

*Figure 2 continued*

multiple technical repeats each. N = 15–80 per line. (**C**) FRAP measurements repeated in the presence of 100 µM carbenoxolone (CBX). (**D, E**) FRAP measurements on cells transfected with siRNA to knockdown *GJA1*, *GJB3*, or *GJB2*. Data normally distributed (Kolmogorov–Smirnov test). Statistical test by one-way ANOVA. * denotes significant decrease in permeability relative to scrambled construct control. (**F**) Immunofluorescence in monolayers showing nuclei stained with DAPI (blue) and connexin Cx26 (green), where present. (**G**) Immunofluorescence performed with Cx43 antibody (green). Images ranked by increasing connexin signal at cell-cell contacts. (**H**) Western blot of LOVO and Caco2 sub-populations following FAC-sorting by Cx43-status. (**I**) Confirmation that sgRNA or siRNA against *GJB2* decreases the expression of Cx26 in DLD1 and SW1222 cells. (**J**) siRNA knockdown of *GJB2* eliminates Cx26 immunofluorescence signal at cell-to-cell contacts in SW1222 cells. (**K**) Blot for Cx26 in DLD1 cells and knockout (KO1-3) clones, and confocal image of calcein-loaded monolayers established from wild-type (WT) or KO1 cells. (**L**). Confluent DLD1 *GJB2* KO (clone 1) monolayers had substantially reduced cell-to-ell connectivity, as determined by FRAP. Mean ± SEM of 20 cells from three monolayers for each genotype. Data normally distributed (Kolmogorov–Smirnov test). Statistical test by one-way ANOVA. *** denotes significant difference (p<0.001) from WT.

The online version of this article includes the following source data and figure supplement(s) for figure 2:

**Source data 1.** Full-length scans of blots for *Figure 2H, I and K*.

**Figure supplement 1.** Western blot analysis of the effects of connexin gene ablation on the expression of selected connexin isoforms in DLD1 and C10 cells, including confirmation of knockdown efficacy.

**Figure supplement 1—source data 1.** Full-length scans of blots for *Figure 2—figure supplement 1A–F*.

**Figure supplement 2.** Confirmation of the knockdown efficiency of siRNA construct against *GJA1* (Cx43) in C10 cells.

**Figure supplement 2—source data 1.** Full-length scans of blots for *Figure 2—figure supplement 2*.

Functional estimates of coupling correlated well with levels of Cx26 protein at cell-to-cell contacts (*Figure 2F*). Furthermore, Cx43 was detected at cell-to-cell contacts in C10, LOVO, and Caco2 but not in RKO, SW1222, or HT29 cells (*Figure 2G*). Unlike Cx26 staining, which was uniform among cells, Cx43 immunofluorescence appeared more heterogenous, with only some cells in a monolayer appearing strongly positive. To confirm the co-existence of Cx43-positive and negative sub-populations, LOVO and Caco2 were FAC-sorted by Cx43 status, expanded, and then analyzed by Western blot. This analysis identified distinct Cx43-positive and -negative sub-populations, indicating that not all cells of a nominally *GJA1*-positive line may benefit from Cx43-dependent connectivity (*Figure 2H*).

Ablation of *GJB2* expression by guide RNA (gRNA) or siRNA (*Figure 2I*) eliminated Cx26 immuno-fluorescence (*Figure 2J*). *GJB2* KO clones of DLD1 cells, generated by CRISPR/Cas9 (*Figure 2K*), had dramatically reduced diffusive exchange (*Figure 2L*). *GJB2* knockdown did not affect Cx31 or Cx43 expression patterns in DLD1 cells (*Figure 2—figure supplement 1*), arguing against compensatory transcriptional responses. In summary, Cx26 channels provide a principal route for cell-to-cell communication in most CRC lines, and the level of this protein is expressed relatively uniformly among cells.

## Small cytoplasmic molecules equilibrate across coupled CRC monolayers

FRAP-based measurements can identify the Cx channels that underpin permeability but cannot predict the extent to which solutes equilibrate at steady state. The latter was evaluated from the degree of fluorescent dye exchange in co-cultures prepared from two populations of cells loaded with spectrally distinct CellTracker fluorophores. The diffusive properties of CellTracker dyes, determined by FRAP (*Figure 3A*), correlated inversely with molecular weight (*Figure 3B*). CellTracker Violet and Orange were selected for co-culture experiments (*Figure 3C*). Monolayers were imaged sequentially for Violet or Orange in confocal mode. In control monolayers prepared with one dye only, fluorescence bleed-through between channels was minimal (*Figure 3D*). To aid visualization, CellTracker Violet and Orange fluorescence images were pseudocolored as blue and red, respectively, such that pixels containing both dyes (i.e., following diffusive exchange) appear purple (*Figure 3D*). In eight CRC lines spanning a range of *GJB2* expression, dye exchange was confirmed, particularly in areas of high cell confluency (*Figure 3E*) which points to a role of cell-to-cell contacts in exchange. As confirmation that dye mixing was due to exchange across cell-to-cell contacts, co-culture experiments were repeated on cells seeded at very low density, which reduces the incidence of cell-to-cell contacts. As expected, a lower incidence of cell-cell contacts resulted in reduced dye-exchange (*Figure 3F*). Furthermore, connexin-negative RKO co-cultures did not show evidence for significant mixing between cells (*Figure 3F*).

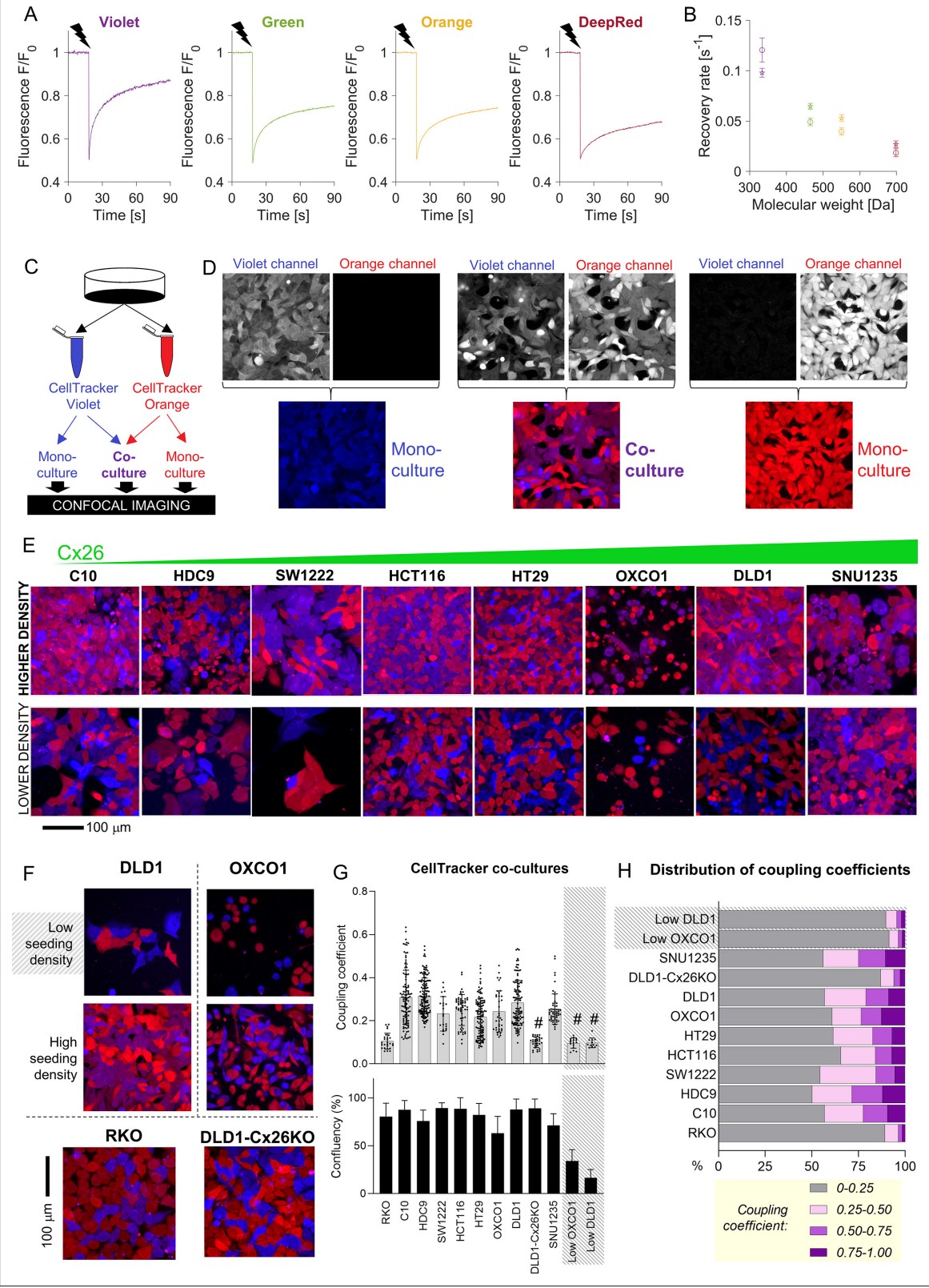

**Figure 3.** Fluorescent molecules equilibrate between coupled cells: imaging. (**A**) Representative time courses of fluorescence recovery after photobleaching (FRAP) protocol for measuring cytoplasmic diffusivity of CellTracker dyes in DLD1 cells in sparse culture. (**B**) Mean recovery rate constant as a function of the molecular weight of the CellTracker dye. Mean ± SEM; N = 20–25 DLD1 cells (star symbol), 7–15 LOVO cells (circles). (**C**) Schematic for preparing co-cultures or mono-cultures loaded with CellTracker dyes. (**D**) Confocal imaging of DLD1 monolayers. CellTracker Orange

*Figure 3 continued on next page*

*Figure 3 continued*

is pseudocolored red and CellTracker Violet is pseudocolored blue; mixing produces purple appearance. (**E**) Images of co-cultures from eight colorectal cancer (CRC) lines, ranked by increasing *GJB2* message, acquired from confluent and low-confluency regions. (**F**) Exemplar images of monolayers established from high- and low-density cultures, showing more exchange where cells make extensive cell-to-cell contacts. For comparison, RKO cells shown at high density; these cells do not express connexin genes. (**G**) Quantification of dye exchange in terms of coupling coefficient. Note that mono-cultures produce no (Violet) or very low (Orange) coupling coefficients (see *Figure 3—figure supplement 1*). Mean ± SEM of multiple fields of view from 4 to 7 biological repeats. Shaded data indicate data from 'low' seeding density, that is, low coupling. # denotes significant (p<0.05) difference between low and high seeding density, performed pairwise for OXCO1 and DLD1 cells. (**H**) Frequency distribution of coupling coefficient ranked by decreasing coupling strength.

The online version of this article includes the following source data and figure supplement(s) for figure 3:

**Source data 1.** Permeability data from FRAP experiments.

**Source data 2.** Coupling coefficient data from CellTracker exchange experiments.

**Figure supplement 1.** Confirmation that mono-cultures established using one CellTracker dye only produce negligible coupling coefficients, as expected from the absence of a second dye.

The degree of dye-mixing arising from cell-to-cell exchange was quantified in terms of a coupling coefficient. This pixelwise analysis measures the fluorescence emitted from each of the two Cell-Tracker dyes as a fraction of total fluorescence and reports the lower of these two numbers as the coupling coefficient (see *Supplementary file 1* for details). Thus, before dye exchange takes place, the coupling coefficient is zero; if dye exchange takes place across gap junctions, the coefficient increases as cells become dually fluorescent. In mono-cultures, the coupling coefficient was negligible, as expected from a control experiment in which only one dye is present (*Figure 3—figure supplement 1*). In contrast, co-cultures produced significant coupling coefficients in the case of cells expressing functional gap junctions seeded at adequate density to form close cell–cell contacts (*Figure 3G*). Connexin-negative RKO cells, DLD1 *GJB2* KO cells, or DLD1 and OXCO1 cells plated at low density produced low coupling coefficients. Strikingly, there was no correlation between FRAP-measured calcein permeability and coupling coefficient determined at steady state for CellTracker dyes. Thus, even monolayers expressing low-conductance gap junctions may produce a uniform distribution of diffusible substances at steady state. Coupling coefficients showed considerable heterogeneity when plotted as a frequency distribution, with weakest coupling in RKO cells and low-density DLD1 and OXCO1 monolayers (*Figure 3H*). In the other cell lines tested, around half of the cells showed evidence of significant coupling. This number is likely to underestimate the extent of coupling in tissues, where contacts can be established across three dimensions.

Some degree of dye leakage from cells is inevitable during long-term culture, but this is unlikely to result in the transfer of dye into neighboring cells. Firstly, CellTracker dyes are chemically rendered to be membrane-impermeable inside cytoplasm, thus any molecules that leak across damaged membranes cannot (re)enter neighboring cells. Secondly, the volume of extracellular space is far greater than the intracellular volume, thus any released dye would become vanishingly diluted outside cells. For these reasons, the above dye-exchange measurements report the consequences of cell-to-cell diffusion across gap junctions, rather than transfer across the extracellular milieu.

Dye exchange was tested further using flow cytometry (*Figure 4A*). Co-cultures were prepared using various pairs of spectrally resolvable CellTracker dyes. The gain of detection channels was calibrated using mono-cultures prepared using one dye only. After gating-out doublets, fluorescence was measured on two detection channels. DLD1 cells grown as mono-cultures emitted fluorescence detected on one channel only (*Figure 4B*). Co-cultures, in contrast, produced a cluster of cells emitting fluorescence in both channels, indicating that dye exchange had occurred during culture. This can be visualized by overlaying pseudocolored density histograms for co-cultures (as green) against the two mono-cultures (as red or blue). To confirm that dye exchange had taken place whilst cells were connected as part of an intact monolayer, control experiments mixed two suspensions of cells harvested from distinctly labeled mono-cultures immediately prior to cytometry (i.e., without a period of culture when cells are coupled to one another). These experiments produced two separate clusters (*Figure 4C*), confirming that dual labeling arises only in intact co-cultures. Similar experiments were performed in other CRC cells (*Figure 4D–G*). The extent of dye exchange between coupled cells was calculated to be ~60% (DLD1 or HT29 cells; *Figure 4H*).

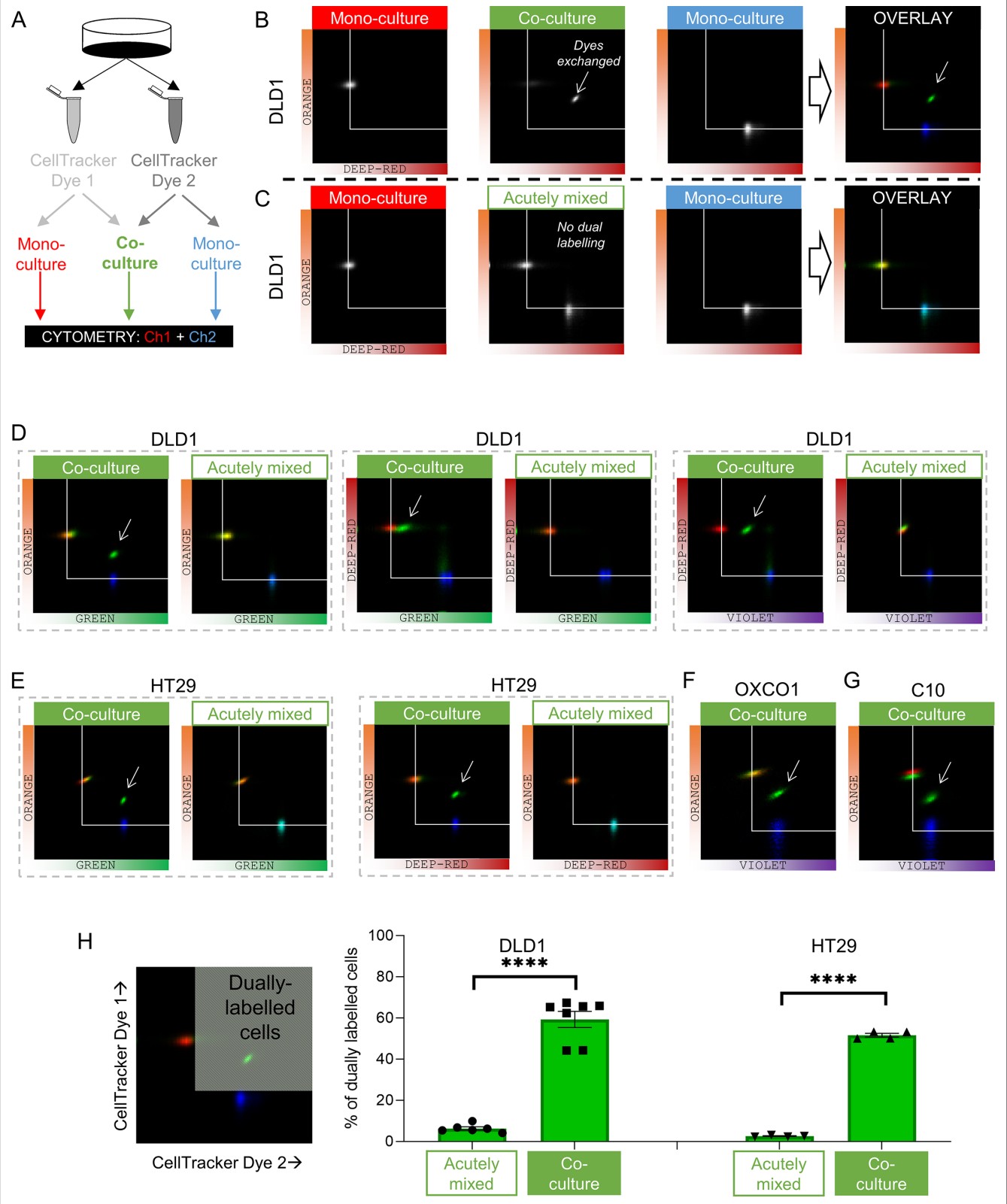

**Figure 4.** Fluorescent molecules equilibrate between coupled cells: cytometry. (**A**) Schematic for producing mono-cultures or co-cultures with pairs of CellTracker dyes. (**B**) Cytometry of DLD1 mono-cultures grown from cells loaded with either DeepRed or Orange, or co-cultures grown from 1:1 mix of cells loaded with DeepRed and Orange. Logarithmic axes showing signal on the relevant detection channels. CellTracker Orange mono-cultures emit signal on the orange detection channel only; CellTracker DeepRed mono-cultures emit signal on the DeepRed detection channel only; co-cultures emit

*Figure 4 continued on next page*

*Figure 4 continued*

fluorescence on both detection channels (indicating that dye exchange had taken place). Overlay shows pseudocolored bivariate density maps: red for Orange mono-cultures, blue for DeepRed mono-cultures and green for co-cultures. Arrow points to cluster showing evidence for dye exchange. (**C**) Flow cytometry of DLD1 cells that were grown as mono-cultures labeled with either DeepRed or Orange. Aliquots of DeepRed- and Orange-labeled cells were mixed prior to flow cytometry to test for acute dye exchange. Analyses show no evidence for dye exchange between acutely mixed cell suspensions. (**D**) Cytometry of co-cultured or acutely mixed DLD1, (**E**) HT29, (**F**) OXCO1, or (**G**) C10 cells loaded with various pairs of CellTracker dyes. (**H**) Quantification of the fraction of cells that had exchanged CellTracker dyes. Region thresholds defined by 95th percentile of signal detected on channels. Significant dye exchange took place only in co-cultured cells but not acutely mixed cells. Mean ± SEM. Significance testing by *t*-test: ****p<0.0001. Each data point represents a separate batch of cells processed for flow cytometry.

The online version of this article includes the following source data for figure 4:

**Source data 1.** Quantification of flow cytometry results.

In summary, coupled CRC cells can freely exchange fluorescent dyes across connexin channels, which argues that small metabolites are expected to equilibrate across the continuous cytoplasmic compartment. The next series of experiments sought evidence that diffusive exchange can functionally rescue cells carrying a genetic inactivation of a specific metabolite-handling process.

## Coupling rescues genetically-inactivated trans-membrane Na$^+$/H$^+$ exchange

The Na$^+$/H$^+$ exchanger NHE1, coded by *SLC9A1*, is a prominent regulator of intracellular pH (pHi), expressed in many CRC cells including HCT116. Genetic inactivation of NHE1 affects pHi control and is detrimental to general cellular physiology (*Hulikova et al., 2013*). If gap junctions were able to conduct a sufficient flow of H$^+$ ions between coupled cells, then it is conceivable for an *SLC9A1*-deficient cell to have its pHi-regulatory needs serviced by a neighboring WT cell. To test this, two *SLC9A1* KO clones were generated using CRISPR/Cas9 (*Figure 5A*). Confirmation that these KO cells lack NHE1 activity was sought in monolayers loaded with the pH-reporter dye cSNARF1. In WT HCT116 cells, NHE1 activity was measured from the rate of pHi recovery following an acid-load (ammonium prepulse) in the absence of CO$_2$/HCO$_3^-$ to eliminate HCO$_3^-$-dependent transporters. NHE1 activity was confirmed from the inhibitory effect of cariporide (*Figure 5B*). KO mono-cultures, in contrast, produced no cariporide-sensitive pHi recovery. To test whether pHi regulation in KO cells could be rescued by coupling onto WT cells, co-cultures were established, wherein KO cells were identified by transfected GFP fluorescence (*Figure 5C*). pHi recovery, following an ammonium prepulse, was tracked simultaneously in WT and KO cells (*Figure 5D*). Strikingly, KO cells showed evidence for pHi recovery when co-cultured with WT neighbors, despite the absence of *SLC9A1* expression. WT cells in co-culture produced slower pHi recoveries than in mono-cultures because of the additional burden of having to service KO cells. Similar observations were made with either KO clone. These experiments illustrate phenotypic blurring: apparent function is more homogenous than the underlying genotype.

To test how diffusive exchange impacts steady-state pHi, measurements were performed on cells equilibrated in HEPES/MES-buffered medium over a range of extracellular pH (pHe). In WT cells, the pHe–pHi relationship was linear and not affected by KO of *GJB2* (*Figure 5E*). Steady-state pHi was reduced in KO mono-cultures, as expected from impaired pHi control. However, the pHi difference between WT and KO cells collapsed in co-cultures. This effect was Cx26-dependent because *SLC9A1*-expressing HCT116 cells that had genetically inactivated *GJB2* were unable to raise the pHi of co-cultured KO cells (*Figure 5E*). Since the readout of pHi regulation was recorded with single-cell resolution, the distribution of pHi among cells was analyzed to seek an effect of coupling on cell-to-cell variation. *SLC9A1*-deficient cells have weaker pHi control, which manifests as a wider pHi distribution in mono-cultures (*Figure 5F*). When, however, these KO cells were co-cultured with WT cells, the pHi distribution returned to normal levels. This observation indicates that NHE1 activity in WT cells was able to service the pHi-regulatory needs of KO cells, manifesting as a narrowing of the population-wide spread of pHi. This rescue effect was Cx26-dependent because knockdown of *GJB2* resulted in a wider pHi spread (*Figure 5F*).

To test whether diffusive exchange benefits the growth of *SLC9A1*-deficient cells, co-cultures were established from different seeding ratios of WT and KO cells. Parallel plates were set-up to record growth terminated at various time points. Total biomass (i.e., growth of WT and KO cells) was measured by the sulforhodamine B (SRB) absorbance assay and growth of the KO compartment

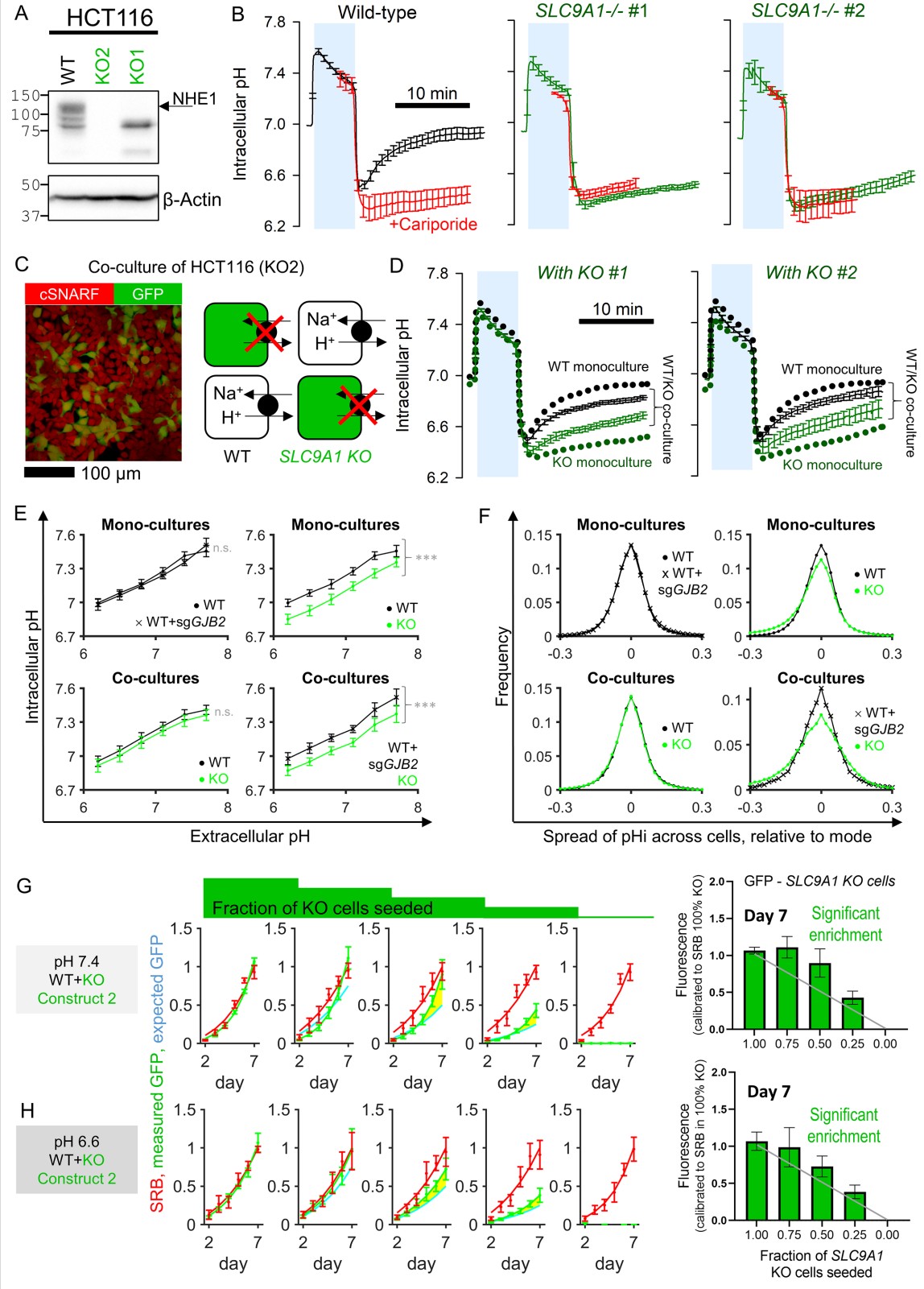

**Figure 5.** Diffusive coupling rescues genetically inactivated Na⁺/H⁺ exchange function. (**A**) Western blot for NHE1, product of *SLC9A1* in wild-type (WT) HCT116 and two knockout (KO) clones. gRNA for KO1 produced a truncated protein, whereas gRNA for KO2 produced complete ablation of expression. (**B**) Functional confirmation of genetic inactivation of *SLC9A1*. Measurements of intracellular pH (pHi) in HCT116 monolayers: following acid-loading by ammonium prepulse (absence of $CO_2/HCO_3^-$ buffer), NHE1 activity in WT (+) cells mediates recovery of pHi (N = 13 biological repeats, with

*Figure 5 continued on next page*

eLife Research article

Figure 5 continued

three technical repeats each), blocked pharmacologically with cariporide (30 μM; N = 6 biological repeats, with three technical repeats each). KO clones (N = 7–9 biological repeats, with three technical repeats each) produce no NHE1 activity. Mean ± SEM. (**C**) Co-culture of WT and KO HCT116 cells. Cells loaded with cSNARF to stain cytoplasm; GFP fluorescence emitted from KO cells. (**D**) Ammonium prepulse performed on co-cultures, separating signal from WT (GFP-negative: black) and KO cells (GFP-positive: green). Mean ± SEM (N = 9–13 biological repeats, with three technical repeats each). For comparison, dotted line shows results from WT or KO mono-cultures. (**E**) Relationship between extracellular and intracellular pH measured in HEPES-MES buffered media (absence of $CO_2/HCO_3^-$ buffer) for WT cells, *GJB2* knockdown cells, or *SLC9A1* KO cells. Results from clones 1 and 2 were not significantly different and pooled together. Measurements were performed for mono-cultures or co-cultures. Mean ± SEM (clockwise: N = 6 WT and 6 WT sg*GJB2* mono-cultures; 6 WT and 7 KO mono-cultures; 8 WT + KO co-cultures; 6 WT sg*GJB2* + KO co-cultures; each with six technical repeats). Statistical testing by two-way ANOVA for pH and genotype; effect of genotype reported in figure (***$p<0.001$). (**F**) Analysis of pHi data from (**E**) in terms of frequency distribution of pHi, offset to the mode. (**G**) Growth curves for HCT116 cells grown as co-cultures of various ratios of WT and KO2 cells, quantified in terms of GFP fluorescence (KO compartment) and sulforhodamine B (SRB) absorbance (total biomass) over 7 days of culture, starting from a seeding density of 2000 cells/well. Mean ± SEM (N = 5 per construct, with four technical repeats each). Significant enrichment indicates that the KO compartment expanded faster than expected from the SRB curve and seeding ratio (2000:0, 1500:500, 1000:1000, 500:1500, and 0:2000), indicating that KO cells benefited from coupling onto WT neighbors. Statistical testing by two-way ANOVA between GFP time course and SRB time course scaled by initial seeding ratio (e.g., 0.5 for 1:1 co-culture); p-value reported for difference between measured growth of KO2 cells (GFP) and prediction growth (SRB, scaled by KO2:WT seeding ratio). Significance at $p<0.05$ indicated by shading between measured (green) and predicted (cyan) time course. Media were at pH 7.4. Bar graph shows GFP signal at day 7 of culture. Gray line is predicted signal. Significant enrichment of GFP cells for seeding ratios of 1:1 and 1:3 (KO2:WT). (**H**) Experiments repeated using media at pH 6.6.

The online version of this article includes the following source data and figure supplement(s) for figure 5:

**Source data 1.** Full-length scans of blots for *Figure 5A*.

**Figure supplement 1.** Growth curves for HCT116 cells grown as co-cultures of various ratios of wild-type (WT) and knockout (KO) 1 cells, quantified in terms of GFP fluorescence (KO compartment) and sulforhodamine B (SRB) absorbance (total biomass) over 7 days of culture, starting from a seeding density of 2000 cells/well.

was determined from GFP fluorescence. In the case of KO mono-cultures, the GFP and SRB signals measure an equivalent quantity and therefore provide a calibration curve for converting between the two types of measurements. When applied to co-culture experiments, this calibration can convert the measured SRB signal into an *expected* GFP signal for comparison against the *measured* GFP signal (i.e. actual growth of the labelled compartment). If WT and KO cells had indistinguishable survival prospects, then the time courses of GFP and SRB should be stoichiometrically related to the seeding ratio and measured GFP should equal expected GFP (i.e. SRB scaled by the seeding density, e.g., by 50% for a 1:1 co-culture). If, however, there is a growth advantage for KO cells in the presence of WT cells, then measured GFP will exceed that predicted from SRB. To test for such rescue, co-cultures were established over a range of seeding ratios. Experiments were performed in media at pH 7.4 or 6.6 to stimulate NHE1 over a range of pH. For co-cultures seeded from 25% KO or 50% KO cells, growth of the GFP-tagged KO compartment exceeded the prediction based on equal WT/KO survival (*Figure 5G and H*). Similar findings were observed using the second KO line (*Figure 5—figure supplement 1*). Thus, co-cultured *SLC9A1*-deficient cells grew faster than expected at the expense of coupled WT cells. Since pHi regulation consumes a substantial amount of energy, this effect could be explained by the exploitation of WT resources by KO cells.

In summary, we show that cells lacking an important ion exchanger at the membrane can hijack the equivalent activity from neighboring WT cells, provided that the relevant ions are able to diffuse across connexin channels.

## Coupling rescues genetically inactivated glycolytic metabolism

The next test related to glycolysis, a critically-important metabolic pathway that handles small molecules in cytoplasm. DLD1 cells were selected for these experiments based on their high glycolytic rate. A recent whole-genome CRISPR-Cas9 screen identified *ALDOA*, a gene coding for the glycolytic enzyme aldolase A, as essential for CRC cell survival under physiological pH (*Michl et al., 2022*). Ablation of *ALDOA* using virally transduced gRNAs reduced the expression (*Figure 6A*) and glycolytic rate, assayed in terms of medium acidification (*Figure 6B*) and lactate production (*Figure 6C*). It was not possible to produce a stable *ALDOA* KO clone, ostensibly because of the enzyme's critical role for survival, thus experiments were performed within 6 days after transduction with one of the two gRNA constructs.

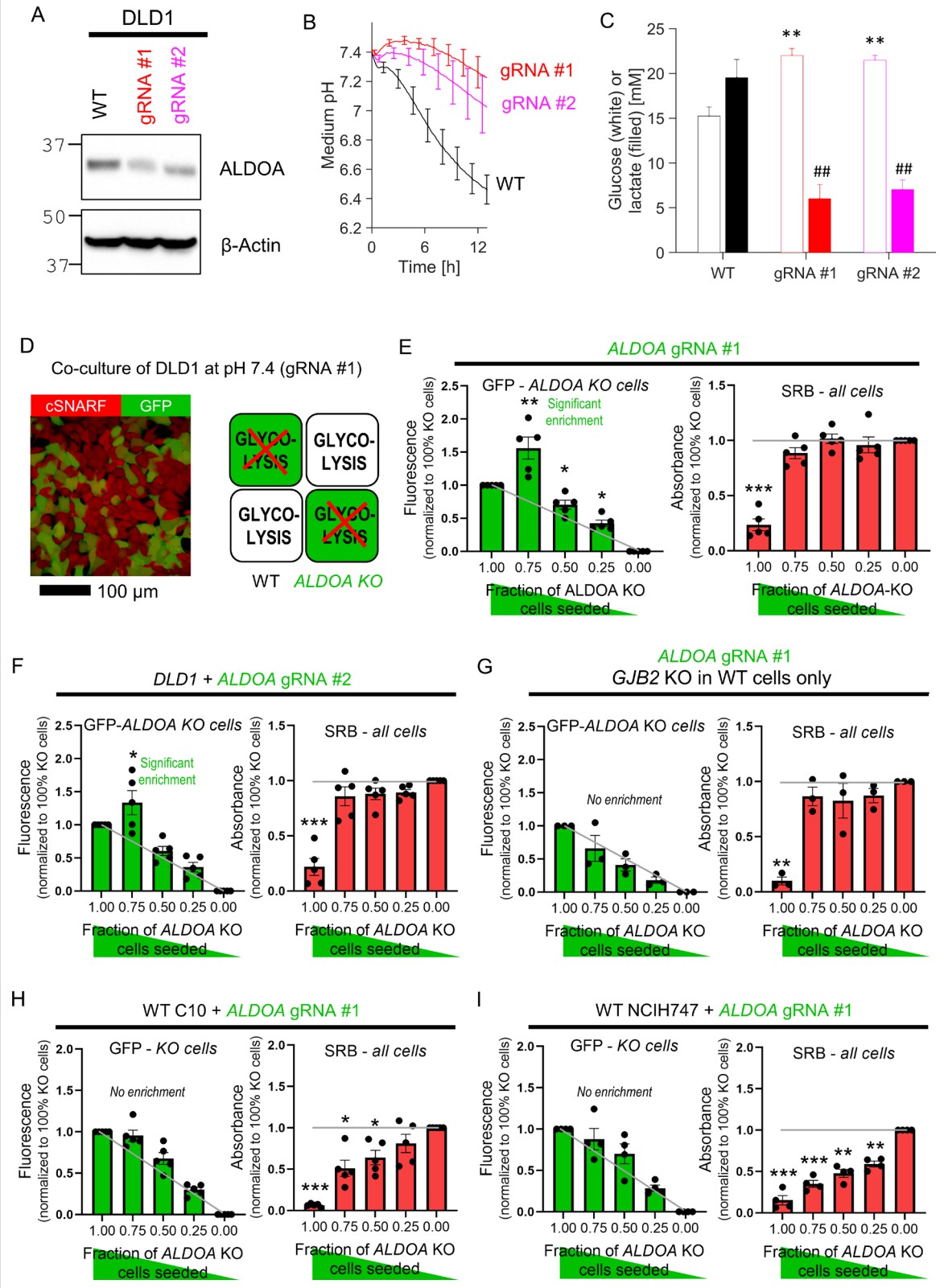

**Figure 6.** Diffusive coupling rescues genetically inactivated glycolysis. (**A**) Western blot for aldolase A (ALDOA) in wild-type (WT) DLD1 cells and cells infected with one of two guide RNA constructs to genetically inactivate *ALDOA*. (**B**) Medium acidification measured in confluent DLD1 cells in low-buffer power media (2 mM HEPES+MES). WT cells are highly glycolytic, as measured from the acidification time course. Glycolytic rate is greatly reduced by *ALDOA* ablation. Mean ± SEM (N = 4 plates, each with three technical repeats). (**C**) DLD1 cells cultured for 4 days in $CO_2/HCO_3^-$ buffered

*Figure 6 continued on next page*

*Figure 6 continued*

media. Media collected at endpoint were assayed for glucose and lactate. Mean ± SEM (N = 3 plates, each three technical repeats pooled per assay). ** denotes significant difference (p<0.01) in glucose consumption relative to WT; ## denotes significant difference (p<0.01) in lactate production relative to WT; testing by one-way ANOVA and multiple-comparisons test. Normality test by Kolmogorov–Smirnov test. (**D**) Co-culture of WT and *ALDOA*-deficient DLD1 cells. Cells loaded with cSNARF-1 to indicate cytoplasm; GFP fluorescence emitted from gRNA-infected cells. (**E**). GFP fluorescence and sulforhodamine B (SRB) absorbance after 6 days of culture of various seeding ratios of WT and *ALDOA*-deficient cells infected with construct #1: 2000:0, 1500:500, 1000:1000, 500:1500, and 0:2000. Left: gray line shows expected GFP signals if growth of WT and *ALDOA*-deficient cells was no different. Right: gray line shows expected SRB signal if growth of WT and *ALDOA*-deficient cells was no different. Significance testing by *t*-test relative to gray line. Mean ± SEM (N = 5 plates, six technical repeats each). Normality test by Kolmogorov–Smirnov test. Significance testing by one-sample *t*-test relative to expected value (shown as gray line). (**F**) Experiment repeated with construct 2. Mean ± SEM (N = 5 plates, six technical repeats each). (**G**) Experiment repeated using *GJB2* knockout DLD1 cells in place of WT DLD1 cells. This substitution has the effect of preventing connexin-dependent coupling between glycolytic DLD1 and *ALDOA*-deficient cells. Mean ± SEM (N = 3 plates, six technical repeats each). (**H**) Co-culture experiment of WT C10 cells with GFP-labeled *ALDOA*-deficient C10 cells (gRNA #1). Mean ± SEM (N = 5 plates, six technical repeats each). (**I**) Co-culture experiment of WT NCIH747 cells with GFP-labeled *ALDOA*-deficient NCIH747 cells (gRNA #1). Mean ± SEM (N = 4 plates, six technical repeats each).

The online version of this article includes the following source data and figure supplement(s) for figure 6:

**Source data 1.** Full-length scans of blots for *Figure 6A*.

**Figure supplement 1.** Sulforhodamine B (SRB) absorbance after 6 days of culture of *ALDOA*-deficient cells infected with construct #1 or #2, normalized to time-matched wild-type (WT) cells.

**Figure supplement 2.** GFP fluorescence after 6 days of culture of various seeding ratios of wild-type (WT) RKO cells with *ALDOA*-deficient DLD1 cells: 2000:0, 1500:500, 1000:1000, 500:1500, and 0:2000.

**Figure supplement 3.** Western blot confirmation of the effect of siRNA knockdown of *ALDOA* on protein levels n (**A**) C10 and (**B**) NCIH747 cells.

**Figure supplement 3—source data 1.** Full-length scans of blots for *Figure 6—figure supplement 3A and B*.

*ALDOA*-deficient DLD1 cells grew four times slower than WT cells (*Figure 6—figure supplement 1*), most likely due to compromised provision of glycolytic ATP and build-up of fructose-1,6-bisphosphate (the substrate for aldolase A). These metabolites are expected to diffuse freely between coupled cells, thus any fructose-1,6-bisphosphate that accumulates in an *ALDOA*-deficient cell should diffuse to neighboring cells for processing, whereas ATP will diffuse in the opposite direction. Diffusive rescue would allow *ALDOA*-deficient cells in co-culture with WT cells to grow faster than expected. To test this, co-culture experiments were performed for various ratios of WT and GFP-labeled *ALDOA*-deficient cells (*Figure 6D*). After 6 days of culture at pH 7.4, GFP fluorescence (a measure of *ALDOA*-deficient cells) and SRB absorbance (total cell biomass) were recorded. If *ALDOA*-deficient cells received no benefit from being coupled onto WT cells, then the endpoint GFP signal would be proportional to the initial seeding density (*Figure 6E*, gray line). However, GFP-labeled cells became relatively enriched in co-cultures with WT cells, which was most prominent at 3:1 seeding ratio. Thus, the survival of *ALDOA*-deficient cells improved when these connected onto WT counterparts. A similar effect was seen using the other gRNA (*Figure 6F*). Rescue of *ALDOA*-deficient cells in co-culture with WT cells was absent when *GJB2* was knocked out in WT cells with gRNA (*Figure 6G*), indicating a critical role for Cx26-mediated metabolite exchange in this rescue. As a negative control, co-culturing *ALDOA*-deficient DLD1 cells with WT RKO cells, a line that does not support significant connexin connectivity, failed to rescue *ALDOA*-deficient cells (*Figure 6—figure supplement 2*).

To test whether connexin isoforms other than Cx26 are able to support metabolic rescue, experiments were performed on two *GJB2*-low cell lines: C10 and NCIH747. C10 cells express Cx43 channels, which produce good cell-to-cell coupling, albeit in a sub-population of cells due to its heterogeneous expression pattern (see *Figure 2G*). Co-culturing of *ALDOA*-deficient C10 cells with WT counterparts did not result in significant metabolic rescue (*Figure 6H*; see *Figure 6—figure supplement 3* for knockdown confirmation). This lack of large-scale rescue may relate to the apparent heterogeneity of Cx43 levels across cells, which allows only Cx43-positive cells to benefit from metabolic rescue. NCIH747 cells express isoforms Cx31 and Cx45 but neither were unable to rescue *ALDOA*-deficient cells in co-culture with WT cells (*Figure 6I*; see *Figure 6—figure supplement 3* for knockdown confirmation).

In summary, we demonstrate that the genetic ablation of a metabolite-handling cytoplasmic enzyme in one cell can be rescued by diffusive access to the required catalysis in neighboring cells via connexin channels. Thus, despite major differences in genotype, the apparent phenotype of KO

and WT cells is more similar. Cx26-assembled channels emerge as particularly effective in supporting metabolic rescue.

## Coupling rescues genetically-inactivated mitochondrial respiration in vitro

The third test for metabolite exchange used cells with genetically inactivated mitochondrial respiration on the basis that this organellar process influences the levels of diffusible substances in cytoplasm, including ATP. SW1222 cells were selected for this experiment because of their high respiratory rate (*Figure 7B*) that could be genetically inactivated by knockout of *NDUFS1*, a gene coding for a component of complex I (*Figure 7A*; *Michl et al., 2022*). Oxidative phosphorylation was blocked in *NDUFS1* KO cells, and a compensatory increase in glycolytic rate ensued (*Figure 7B*). Cells deficient in *NDUFS1* grew three times slower than WT cells (*Figure 7—figure supplement 1*), likely due to ablated mitochondrial metabolism that is critical for delivering substrates and energy. To test whether junctional coupling could rescue *NDUFS1*-deficient cells, WT and GFP-labeled KO cells were co-cultured in medium at pH 7.7 (i.e. alkaline conditions that activate glycolysis fully; *Figure 7C*). If KO cells received no benefit from being coupled onto WT cells, the GFP signal measured at endpoint would be proportional to the seeding ratio. However, co-culture with WT cells promoted the growth of GFP-labeled KO cells beyond the expected level, particularly for seeding ratios of 3:1 KO:WT (*Figure 7D*). This rescue related to the density of cell-to-cell contacts because KO cells could no longer receive a growth benefit in sparse co-culture with WT cells (*Figure 7E*). Furthermore, the rescue was linked to Cx26-dependent coupling as co-culture with WT cells that had genetically knocked down *GJB2* by siRNA yielded no benefit to co-cultured *NDUFS1*-deficient cells (*Figure 7F*). Ultimately, the rescue effect of WT cells relates to oxidative phosphorylation, which requires oxygen and should be lessened under hypoxic conditions. Indeed, this was the case in co-cultures incubated in 2% $O_2$ (*Figure 7G*).

The importance of Cx26-assembled channels in providing metabolic rescue to *NDUFS1* KO cells was tested in co-cultures with *GJB2*-low/negative lines: RKO, C10, and NCIH747 (*Figure 7H*). Connexin-negative RKO cells were unable to rescue *NDUFS1*-deficient SW1222 cells. *GJA1*-positive C10 cells, which form Cx43-dependent connections, did not rescue *NDUFS1*-deficient cells, indicating that Cx26 and Cx43 channels are unlikely to form heterotypic channels (consistent with previous findings *Koval et al., 2014*). Finally, NCIH747, which produce Cx31 and Cx45, also did not rescue *NDUFS1*-deficient cells, further supporting the notion that heterotypic channels are less efficacious than Cx26-assembled conduits.

## Cx26-dependent coupling rescues genetically-inactivated mitochondrial respiration in vivo

Whilst in vitro assays provide a well-controlled environment for evaluating metabolic rescue, their findings must be verified in vivo, wherein cells are able to connect extensively in three dimensions and grow over many cycles of division. Evidence for connexin-dependent metabolic rescue in vivo was tested in xenografts seeded from a 1:1 mixture of GFP-labeled *NDUFS1*-deficient (KO) SW1222 cells and DLD1 cells. In this system, DLD1 cells lay down a cellular network for *NDFUS1* KO cells to grow in. If metabolic rescue took place, then actively respiring DLD1 cells would promote the growth of the mitochondrially-defective (GFP-labelled) cellular population. Xenografts of WT DLD1 plus *NDUFS1* KO SW1222 cells were established subcutaneously on the left flank of mice. As a control, the opposite flank of the mouse was injected with a mixture of *NDUFS1* KO SW1222 cells plus Cx26 knockout (*GJB2* KO) DLD1 cells (*Figure 8A*), that is, a combination that cannot establish significant strong DLD1-SW1222 coupling. In DLD1 cells, genetic inactivation of *GJB2* had no substantial effect on cell growth (*Figure 8B*) or ability to form 3-D spheroids (*Figure 8C*), thus both types of DLD1 cells were deemed suitable for growing xenografts.

In terms of bulk volume, xenografts grew faster on the left flank, wherein DLD1 and SW1222 cells can establish Cx26-assembled gap junctions (*Figure 8D*). At the endpoint (a combined left and right tumor burden of >1 cm³ or 30 days of growth), xenografts were excised for histology and stained with the nuclear dye Hoechst and GFP antibody to enhance visualization of GFP-labelled *NDUFS1*-deficient cells (*Figure 8E*). GFP-positive cells were identified by an image analysis pipeline (see 'Materials and methods') that quantifies the number of fluorescent clusters and their combined area, a process repeated across multiple slices through the tumor height. This analysis confirmed

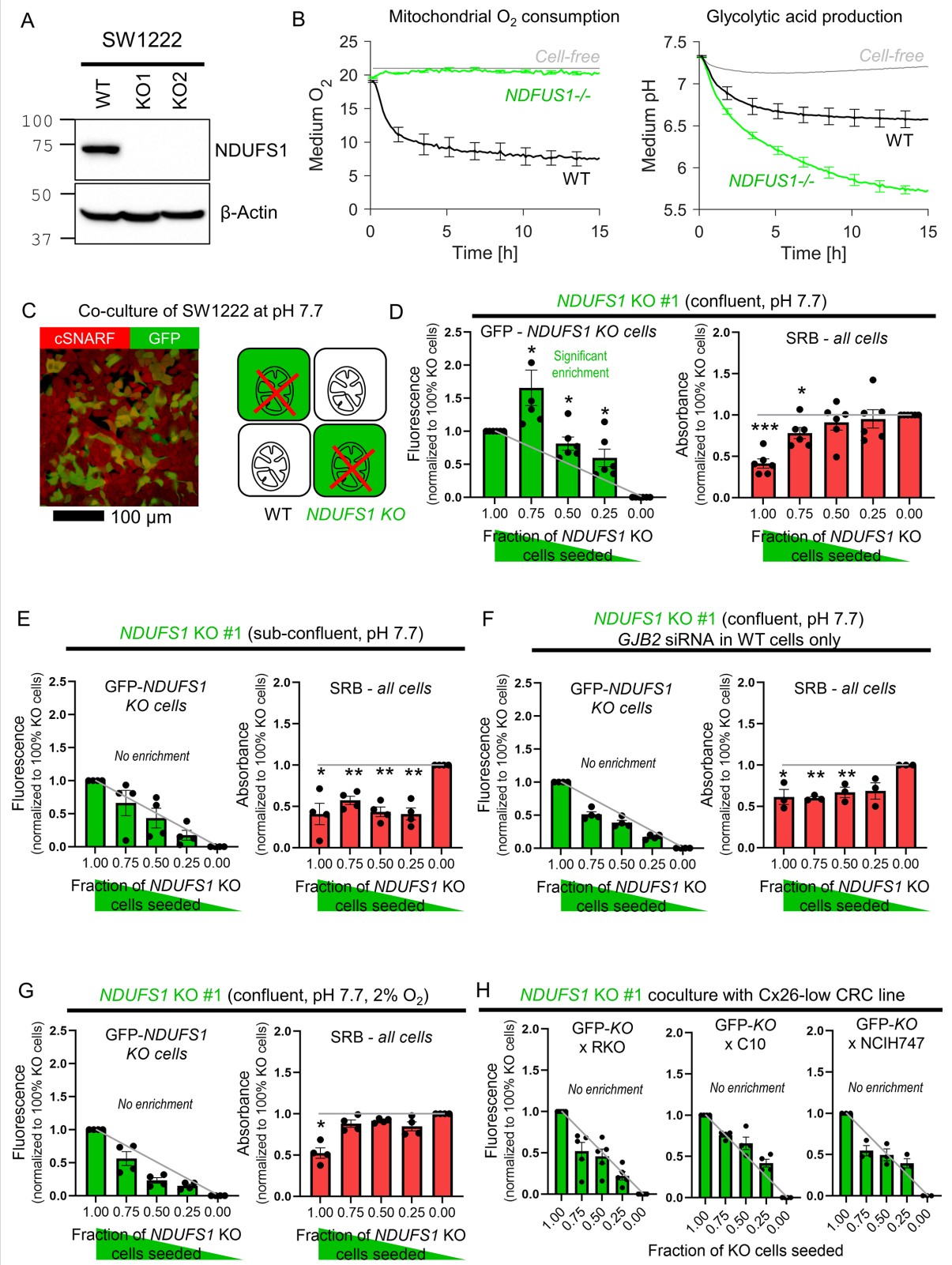

**Figure 7.** Diffusive coupling rescues genetically inactivated mitochondrial respiration. (**A**) Western blot for NDUFS1 in wild-type (WT) SW1222 cells and *NDUFS1* knockout (KO) clones established using one of two guide RNAs. (**B**) Fluorimetric measurements of oxygen consumption and acid production (assays of respiratory and glycolytic rates) in WT and *NDUFS1* KO SW1222 cells. Mean ± SEM (N = 3 plates, three technical repeats each). (**C**) Co-culture of WT and *NDUFS1* KO cells. Cells loaded with cSNARF-1 to indicate cytoplasm; GFP fluorescence emitted from KO cells. (**D**) GFP fluorescence and

*Figure 7 continued on next page*

Figure 7 continued

sulforhodamine B (SRB) absorbance after 6 days of culture of various seeding ratios of WT and *NDUFS1*-deficient cells infected with construct #1: 2000:0, 1500:500, 1000:1000, 500:1500, and 0:2000. Left: gray line shows expected GFP signals if growth of WT and *NDUFS1*-deficient cells was no different. Right: gray line shows expected SRB signal if growth of WT and *NDUFS1*-deficient cells was no different. Media pH set to 7.7. Significance testing by *t*-test relative to gray line. Mean ± SEM (N = 4 plates, four technical repeats each). Normality test by Kolmogorov–Smirnov test. Significance testing by one-sample *t*-test relative to expected value (shown as gray line). (**E**) Experiment repeated at lower (1000/well) seeding density to reduce incidence of cell coupling. Mean ± SEM (N = 4 plates, six technical repeats each). (**F**) Experiment repeated with WT cells *GJB2* KD via siRNA, that is, inactivating connexin coupling between WT and *NDUFS1*-deficient cells. Mean ± SEM (N = 3 plates, six technical repeats each). (**G**) Experiment repeated under hypoxic conditions, which suppresses mitochondrial function. Mean ± SEM (N = 4 plates, six technical repeats). (**H**) Co-culture of GFP-labeled *NDUFS1* KO cells (#1) with one of three Cx26-negative cells: WT RKO, C10, or NCIH747 cells.

The online version of this article includes the following source data and figure supplement(s) for figure 7:

**Source data 1.** Full-length scans of blots for *Figure 7A*.

**Figure supplement 1.** Sulforhodamine B (SRB) absorbance after 6 days of culture of *NDUFS1*-deficient cells, normalized to time-matched wild-type (WT) cells (*t*-test).

larger tumors on the left flank that contained a greater number of GFP-positive clusters and a larger combined GFP-positive area (*Figure 8F*). These inferences were unaltered by changing the threshold for defining GFP-positive areas (*Figure 8—figure supplement 1*). Overall, these in vivo findings confirm that DLD1 cells are able to rescue *NDUFS1*-deficient SW1222 cells in a Cx26-dependent manner, consistent with in vitro co-culture observations.

In summary, we show that genetically-inactivated mitochondrial metabolism in one cell can be rescued by coupling onto a WT cell in vitro as well as in vivo. These findings demonstrate that phenotypic differences between cancer cells can be reduced ('blurred') by diffusive coupling (*Figure 8G and H*).

## Discussion

The results of this study show that CRC cells establish a multicellular cytoplasmic continuum through which small molecules are able to diffuse and, with time, equilibrate. Solute exchange was demonstrated using fluorescent dyes in terms of an apparent permeability constant and degree of color mixing at steady state. The apparent permeability to calcein, as measured by FRAP, varied among CRC cell lines and correlated best with *GJB2* expression and Cx26 immunoreactivity at cell-to-cell contacts. The role of this particular connexin isoform in producing functional conduits was inferred from the effect of genetic knockdown on permeability, which was most sensitive to siRNA against *GJB2*. The gene coding for Cx26 is expressed in normal colorectal epithelium (*Kanczuga-Koda et al., 2005*), and the unimodal distribution of its message level among CRC cell lines suggests that transformed cells retain its expression, with a few notable exceptions (e.g., RKO). In addition to *GJB2*, some CRC cells express isoforms *GJA1* and *GJC1*, possibly due to mutations or stable epigenetic changes present only in a subset of lines. Intriguingly, the magnitude of calcein permeability measured by FRAP did not predict the degree to which solutes equilibrate between cells at steady state, as measured by coupling coefficient. This is because even weakly coupled confluent monolayers can exchange dyes, if allowed sufficient time. Thus, the phenomenon of solute equilibration across coupled cellular networks is likely to be significant even in cancer cells with nominally low connexin expression.

In the context of cancer, diffusion through connexin-assembled channels is significant for growth when its traffic includes important metabolites handled by enzymes or transporters. In the event of a loss-of-function mutation, the absence of activity could be compensated by access to fully functional proteins in neighboring cells (*Figure 8G and H*). This rescue effect was confirmed for three cases of biologically important processes that handle small molecules at distinct subcellular domains: at the plasma membrane (NHE1, coded by *SLC9A1*, handling intracellular H$^+$ ions), in the cytoplasm (aldolase A, coded by *ALDOA*, handling the glycolytic intermediate fructose-1,6-bisphosphate), and in mitochondria (NADH:ubiquinone oxidoreductase core subunit S1; *NDUFS1*, part of complex I). Mono-cultures of genetically altered cells manifested a functional defect, namely, impaired pHi regulation (*SLC9A1*-deficient cells), blocked glycolysis (*ALDOA*-deficient cells), or inactivated mitochondrial respiration (*NDUFS1*-deficient cells). However, co-culture with WT counterparts provided rescue, detected as an enrichment in cells carrying the genetic ablation (i.e. the GFP-labelled compartment of

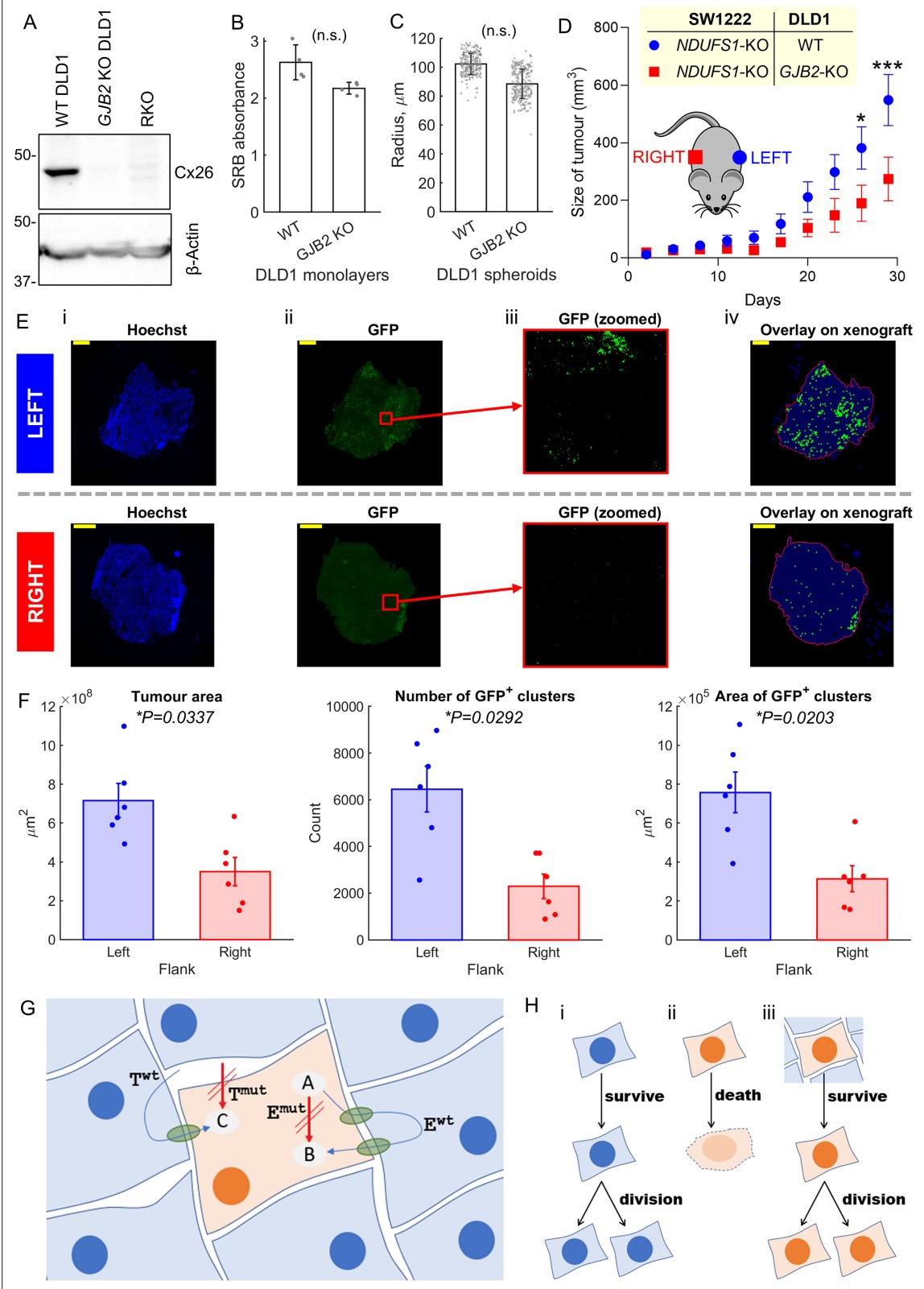

**Figure 8.** Metabolic rescue by connexins channels. (**A**) Western blot for Cx26, showing absence of protein in *GJB2* knockout (KO) DLD1 cells. RKO used as negative control. (**B**) Effect of Cx26 (*GJB2*) knockout on DLD1 cell growth in 2-D monolayer (n = 4; not significant, paired *t*-test). Seeded at 2000/well, growth after 6 days measured by sulforhodamine B (SRB) assay (absorbance, 520 nm). (**C**) Effect of Cx26 (*GJB2*) knockout on ability of DLD1 cell to form spheroids (n = 3 platings, each yielding 60–80 spheroids; not significant, nested paired *t*-test). (**D**) Growth curves of xenografts on left and

*Figure 8 continued on next page*

*Figure 8 continued*

right flank of nude female mice (n = 6). Analysis by repeated-measures two-way ANOVA (*p<0.05, ***p<0.001). Time courses aligned to endpoint. (**E**) Histology of selected 5 μm slices through the middle of a xenograft on the left or right flank (matched animal). Panels show (i) staining for Hoechst, (ii) GFP signal from antibody, (iii) GFP signal in a central area of the field of view, after removing background (thresholding), and (iv) analysis of GFP-positive clusters (indicated by green dot) within outline (red) of tumor mass. Scale bar is 1 mm. (**F**) Results of analysis of left and right flank xenografts from six mice in terms of tumor area, number of GFP-positive clusters, and ensemble area of GFP-positive clusters. (**G**) Cartoon summary of the role of connexin channels, particularly Cx26 (green conduits) in rescuing cell carrying mutations in an enzyme (E$^{mut}$) and membrane transporter (T$^{mut}$). Itself, the mutation-bearing cell is unable to convert substrate A to B (by enzyme E) and take up substance C. However, these functions can be rescued by diffusive exchange of A, B, and C with neighboring cells that express wild-type (WT) T$^{wt}$ and E$^{wt}$. (**H**) Scenario *i*: a WT cell survives and divides. Scenario *ii*: in isolation or in mono-culture, a mutation-bearing cell cannot survive and fails to divide and therefore carry on its genetic mutation. In a human tumor, this would lead to negative selection of such loss-of-function mutations in important genes (coding for E and T). Scenario iii: coupling onto WT cells can rescue the metabolic defect of the mutation-bearing cell, hence allowing survival. These mutations will carry on to daughter cells and therefore not lead to negative selection.

The online version of this article includes the following source data and figure supplement(s) for figure 8:

**Source data 1.** Full-length scans of blots for *Figure 8A*.

**Figure supplement 1.** Analysis of xenograft histology using three levels of threshold to define GFP-positive areas.

co-cultures). Metabolic rescue by connexins was observed in vitro, as well as in vivo, where it allowed a genetically defective cell (*NDFUS1* KO) to expand faster than otherwise. The rescue effect was related to Cx26 channels because it was diminshed by genetically inactivating *GJB2*. However, a role for other connexin isoforms cannot be excluded, although the findings herein suggest that Cx26-low/negative cells expressing *GJB3* (Cx31) or *GJA1* (Cx43) were unable to match the metabolic rescue provided by Cx26 pathways. In the case of Cx31, the conductance established by these channels is likely to be small as its knockdown did not reduce ensemble permeability. Strong conductances could be established in at least some cells of nominally *GJA1* (Cx43)-positive lines. Intruigingly, Cx43-dependent connectivity tended to be heterogenous within a monolayer, which has the consequence of restricting the benefit of metabolic rescue to only a subset of cells. This pattern of expression may explain why Cx43 is less efficacious in metabolic rescue globally, as compared to the more uniformly distributed Cx26.

Metabolic rescue by gap junctions is relevant to our understanding of somatic evolution in cancer because it provides a mechanism by which cells carrying loss-of-function mutations in metabolite-handling genes are able to survive and divide as long as connections are made onto WT neighbors (*Figure 8G and H*). Consequently, cells acquiring such spontaneous mutations will evade negative selection in a human tumor. However, the same mutation studied in a mono-culture may be lethal because this in vitro setting does not offer a means of rescue by wild-type proteins. Metabolic rescue by gap junctions may explain the paradox of why some genes are found to be essential in vitro but not in vivo (i.e., their inactivating mutations are not negatively selected in human cancers). In contrast, genes responsible for processes that do not handle diffusible solutes cannot benefit from connexin-mediated coupling, irrespective of their cellular neighborhood. This category includes genes coding for ribosomal subunits that are too large to cross connexin channels, or genes coding for neoantigens that are, by design, confined to the mutation-carrying cell. Inactivating mutations in such genes may be selected negatively as there are no tangible ways for compensating the ensuing functional deficit (*Zapata et al., 2018*).

Another implication of our findings in the context of carcinogenesis relates to the definition of the 'unit' that is being selected during somatic evolution. A key assumption of current tumorigenesis models is that mutations act in a cell-autonomous fashion. Exceptions to this paradigm have been postulated for autocrine interactions (*Tomlinson and Bodmer, 1997*), but the role of gap junctional connectivity has not been explored. In terms of genotype, each cell in a tumor is a distinct unit, responding independently to selection pressures. However, diffusive coupling between cells of contrasting genotype can produce an emergent phenotype that is less distinct. This scenario can arise in the case of spontaneous mutations affecting metabolite-handling genes and is significant because the coupled cells will have comparable survival prospects, despite striking differences in genotype. In this system, the unit under selection extends beyond a single cell. Our findings emphasize the importance of measuring phenotypic variation, particularly in instances relating to diffusible molecules, as this may not overlay with the more compartmentalized genotypic landscape. In cases

where cell-to-cell phenotypic differences are blurred as a result of connexin connectivity, it would not be appropriate to consider a single cell as the unit under selection (*Bertolaso and Dieli, 2017*). Intriguingly, connexin genes are rarely mutated in cancer (*Zapata et al., 2018*) and we speculate that this reflects a protective role of cell-to-cell coupling in cancer. Therapeutic interventions that target connexins may expose vulnerabilities in cancer and increase the efficacy of treatments, particularly those with demonstrably strong in vitro effects.

# Materials and methods

## Key resources table

| Reagent type (species) or resource | Designation | Source or reference | Identifiers | Additional information |
|---|---|---|---|---|
| Chemical compound, drug | Anti-adherence rinsing solution | STEMCELL Technologies | 07010 | |
| Chemical compound, drug | SULPHORODAMINE for SRB Assay | Sigma-Aldrich | 3520-42-1 | |
| Chemical compound, drug | cSNARF1 | Thermo Fisher | C1271 | |
| Antibody | Anti-Cx31 (mouse monoclonal) | Proteintech | 12880-1-AP | 1:1000 |
| Antibody | Anti-Cx26 (mouse monoclonal) | Invitrogen | 33-5800 | 1:1000 |
| Antibody | Anti-Cx26 (mouse monoclonal) | Proteintech | 14842-1-AP | |
| Antibody | Anti-Cx43 (mouse monoclonal) | Cell Signalling Technology | 3512 | |
| Antibody | Anti-Cx43 (rabbit polyclonal) | Cell Signalling Technologies | 13-8300 | 1:1000 |
| Antibody | Anti-GAPDH (mouse monoclonal) | Proteintech | HRP-60004 | 1:6000 |
| Antibody | Anti-beta actin (mouse monoclonal) | Proteintech | HRP-60008 | 1:6000 |
| Antibody | Anti-Cx43 APC-conjugated (mouse monoclonal) | R&D Systems | FAB7737A | 1:500 |
| Antibody | Anti-NDUFS1 (rabbit polyclonal) | Thermo Fisher | PA5-22309 | 1:3000 |
| Antibody | Anti-aldolase A (rabbit polyclonal) | Novus Biologicals | NBP1-87488 | 1:3000 |
| Antibody | Anti-GFP (rabbit polyclonal) | Thermo Fisher | A-11122 | 1:1000 |
| Chemical compound, drug | Cariporide | Tocris | 5358 | |
| Chemical compound, drug | Lipofectamine RNAiMAX | Invitrogen | 2373383 | |
| Sequence-based reagent | siRNA | Dharmacon | siGENOME SMARTpool M-011042-01-0005 | Silencer Select |
| Sequence-based reagent | siRNA | Dharmacon | siGENOME SMARTpool L-019285-00-0005 | Silencer Select |
| Sequence-based reagent | siRNA | Dharmacon | siGENOME SMARTpool M-019948-02-0005 | Silencer Select |

*Continued on next page*

*Continued*

| Reagent type (species) or resource | Designation | Source or reference | Identifiers | Additional information |
|---|---|---|---|---|
| Sequence-based reagent | siRNA | Dharmacon | siGENOME NONtargeting control D-001210-01-05 | Silencer Select |
| Chemical compound, drug | CellTracker Fluorescent Dye Violet | Invitrogen | C10094 | |
| Chemical compound, drug | CellTracker Fluorescent Dye Orange | Invitrogen | C34551 | |
| Chemical compound, drug | CellTracker Fluorescent Dye Green | Invitrogen | C2925 | |
| Chemical compound, drug | CellTracker Fluorescent Dye DeepRed | Invitrogen | 34565 | |
| Sequence-based reagent | LentiCRISPR v.2 NDUFS1 gRNA1 | http://genome-engineering.org/gecko/wp-content/uploads/2013/12/lentiCRISPRv2-and-lentiGuide-oligo-cloning-protocol.pdf | TAGAATGTATGCCTACTTGG | Targeting gRNA sequence |
| Sequence-based reagent | LentiCRISPR v.2 ALDOA gRNA1 | http://genome-engineering.org/gecko/wp-content/uploads/2013/12/lentiCRISPRv2-and-lentiGuide-oligo-cloning-protocol.pdf | CATTGGCACCGAGAACACCG | Targeting gRNA sequence |
| Sequence-based reagent | LentiCRISPR v.2 SLC9A1 gRNA1 | http://genome-engineering.org/gecko/wp-content/uploads/2013/12/lentiCRISPRv2-and-lentiGuide-oligo-cloning-protocol.pdf | GAGCAGGGTGCTGATGACGA | Targeting gRNA sequence |
| Sequence-based reagent | LentiCRISPR v.2 GJB2 gRNA1 | http://genome-engineering.org/gecko/wp-content/uploads/2013/12/lentiCRISPRv2-and-lentiGuide-oligo-cloning-protocol.pdf | GACATAGAAGACGTACATGA | Targeting gRNA sequence |
| Cell line (human) | Colorectal cancer cell lines | Bodmer laboratory, WIMM, Oxford | | |
| Other | Athymic Nude Crl:NU(NCr)-Foxn1nu | Charles River | | |

## Cell lines and culture

Human CRC cell lines were provided by the Bodmer Laboratory and authenticated by a panel of informative SNPs. Cell lines are regularly tested for mycoplasma contamination by antigen test. Cells were cultured using DMEM (Sigma-Aldrich, D7777) supplemented with 10% FBS and 1% PenStrep (10,000 U/mL) at 37°C and 5% $CO_2$. To adjust medium pH, the content of $NaHCO_3$ was varied (*Michl et al., 2019*).

## siRNA transfection

Cells were seeded at a density of 200,000 cells/well in a 6-well plate and transfected with either siRNA (20 nM), for example, for *GJB2*, *GJB3*, and *GJA1*, or with a scramble non-targeted siRNA (Dharmacon, siGENOME smart pool), using Lipofectamine RNAiMAX (Invitrogen, Cat# 2373383). After 72 hr, cells were harvested and seeded for experiments.

## Viral transduction with CRISPR-CAS9 constructs

Gene KOs were made for DLD1, SW1222, and HCT116 cell lines using DMEM, 10% FBS, 1% Pen/Strep. gRNA sequences were cloned into LentiCRISPR v.2 backbone as previously described (http://genome-engineering.org/gecko/wp-content/uploads/2013/12/lentiCRISPRv2-and-lentiGuide-oligo-cloning-protocol.pdf). Two gRNA sequences were cloned for each gene, using sequences below. Concentrated virus aliquots were prepared by the virus production facility at WIMM, University of Oxford. Cells were plated in clear, flat-bottom 6-well plate at a density of 200,000 cells/well and transduced using a 500 µL aliquot of lentivirus carrying the LentiCRISPR v2 construct encoding for a

gRNA sequence targeting one individual gene. Polybrene was added at a concentration of 4 µg/mL. The 6-well plate was incubated for 2 days before puromycin (5 µg/mL) was added for selection, and it was incubated for 3 days before the transduced cells were used for further experiments. Single-cell clones with stable deletion of *NDUFS1* and *SLC9A1* were obtained in SW1222 and HCT116 cells, respectively. Note that for *ALDOA* gRNA-treated DLD1 cells no stable KO clones could be obtained. Therefore, lentivirus pools of KO cells were mixed populations of cells with different genomic edits and unedited cells. All growth curves were performed within 1 week of initial viral infection to avoid hypomorph or unedited cells outcompeting those that have loss-of-function mutations.

| | |
|---|---|
| NDUFS1 gRNA1 | TAGAATGTATGCCTACTTGG |
| NDUFS1 gRNA2 | TCACAAATAGGACAGTCCAA |
| ALDOA gRNA1 | CATTGGCACCGAGAACACCG |
| ALDOA gRNA2 | AATGGCGAGACTACCACCCA |
| SLC9A1 gRNA1 | AGCAGGGTGCTGATGACGA |
| SLC9A gRNA2 | GATGCCAGACCGCAGAACCA |
| GJB2 gRNA1 | GACATAGAAGACGTACATGA |

## Flow cytometric sorting

Cells were seeded on Petri dishes and harvested when confluent. Cells were resuspended in FACS-compatible solution containing 2% FBS in PBS and transferred to tubes for incubating with Ab for Cx43 APC-conjugated (R&D Systems, Cat# FAB7737A) for 45 min at room temperature (RT). Cells were washed twice with FACS solution and sorted immediately thereafter. Positive and negative cells were collected, expanded, and prepared for Western blotting for Cx43.

## Spheroid growth

WT and *GJB2* KO (Cx26 KO) DLD1 spheroids were prepared by the hanging drop method. Then, 20 µL-drops with 400 DLD1 cells were allowed to form spheroids and grow for 72 hr. Spheroids were then collected and transferred into sterile non-TC flat-bottom 96-well plate (Falcon) pretreated with anti-adherence rinsing solution (STEMCELL Technologies, Cat# 07010) and imaged in brightfield setting using Cytation 5 plate reader to calculate radius.

## Immunoblotting

Lysates were prepared with radioimmunoprecipitation assay (RIPA) buffer. Protein concentration in the samples was measured using bicinchoninic acid (BCA) protein assay kit and adjusted using water. Samples were loaded onto a 10% acrylamide gel. The gel was run at 120 V for 90 min. Afterward, membrane transfer was performed at 250 mA for 70 min. Membranes were incubated with primary antibodies against NDUFS1 (Thermo Fisher, Cat# PA5-22309), aldolase A (Novus Biologicals, Cat# NBP1-87488), NHE1 (BD Biosciences, Cat# 611775), β-actin (Proteintech, Cat# HRP-60008), and GAPDH (Proteintech, Cat# HRP-60004) used as a loading control; connexin 26 (Thermo Fisher, Cat# 13-5800), connexin 31 (Proteintech, Cat# 12880-1-AP), connexin 43 (Thermo Fisher, Cat# 13-8300), and HRP-conjugated goat anti-rabbit and anti-mouse secondary antibodies were applied. The membrane was visualized using ECL.

## Immunofluorescence

Cells were grown to 50–80% confluency, fixed with 4% paraformaldehyde in PBS (Pierce; Life Technologies) and permeabilized with 0.2% Triton X-100 in PBS. After blocking with 3% BSA in PBS for 1 hr, cells were incubated with rabbit antibodies against Cx26 (Proteintech, Cat# 14842-1-AP) and Cx43 (Cell Signaling Technology, n.3512) for 1.5 hr at RT. Cells were then washed and incubated with Alexa Fluor 488 secondary antibody (Life Technologies) for 1 hr. Cells nuclei were co-stained with Hoechst

33342 (Life Technologies) applied for 1 min following washing with 1× PBS. The mounting medium used was ProLong Gold Antifade reagent (Invitrogen, Cat# P36930).

## FRAP for measuring coupling

Monolayers in 4-well ibidi imaging slides were loaded with calcein AM (Invitrogen, C1430) for 10 min in HEPES RPMI, replaced to remove unloaded dye. Confluent monolayers were imaged using Zeiss LSM 700 confocal microscope. A cell in the middle of a confluent cluster was selected for bleaching (high-power 488 nm) until fluorescence (488 nm excitation, emission >510 nm) decreased by 50%. Signal was measured in the central cell, neighbors and more remote cells, and normalized to the initial intensity. Recovery was fitted to a monoexponential to calculate calcein permeability (units: μm/min) from product of the geometric perimeter and area of the central cell, divided by the time constant of fluorescence recovery. See Appendix 1 for details of calculation.

## FRAP for measuring diffusivity

Cells were loaded with Violet (BMQC 5 μM, excitation 405 nm, Invitrogen, Cat# C10094), Green (CMFDA 20 μM, excitation 492 nm, Cat# C2925), Orange (CMRA 20 μM, 555 nm, Cat# C34551), or DeepRed (20 μM, 630 nm, Cat# 34565) for 15 min in RPMI HEPES medium. FRAP was limited to a 3 × 3 μm region of cytoplasm.

## Measuring the exchange of CellTracker dyes in terms of coupling coefficient

A suspension of cells was split equally, and each loaded with either CellTracker Violet or Orange for 15 min. Cells were pelleted, washed (PBS), combined in 1:1 ratio, and then seeded onto 4-well imaging slides at 300,000 cells/well. As control, cells were loaded with one type of dye. After 48 hr, monolayers were imaged confocally using sequential acquisition that minimizes bleed-through between channels: 405 nm excitation and emission 490–555 nm for the Violet channel and 555 nm excitation and emission 560–600 nm for the Orange channel. Orange and Violet fluorescence were normalized to the mean signal in paired control. The degree of dye mixing was quantified in terms of a coupling coefficient (CC):

$$CC = 2 \times \min \left( \frac{F_O}{(F_O + F_V)}, \frac{F_V}{(F_O + F_V)} \right)$$

where $F_O$ and $F_V$ are the normalized Orange and Violet fluorescence signals. See Appendix 1 for details of calculation.

## Measuring the exchange of CellTracker dyes by flow cytometry

Cells were loaded with combinations of CellTracker dyes, co-cultured for 48 hr, and processed for flow cytometry (Attune NXT Analyser). Gain-of-detection channels were optimized by analyzing mono-cultures loaded with one dye. Data were presented as pseudocolored bivariate density plots representing mono-cultures with the low-wavelength dye as blue, mono-cultures with the high-wavelength dye as red, and co-cultures as green.

## Measuring NHE1 activity

Cells seeded on ibidi 4-well slides were loaded with 10 μM cSNARF1 (Thermo Fisher, Cat.# C1271, excitation, 555 nm; emission, 580 and 640 nm) for 7 min. Cells were then washed by superfusion with HEPES-buffered normal Tyrode containing (in mM) NaCl (135), KCl (4.5), $CaCl_2$ (2), $MgCl_2$ (1), HEPES (20), glucose (11), pH adjusted to 7.4 with 4 M NaOH heated to 37°C. After 2 min to allow equilibration, the superfusion was switched to ammonium containing Tyrode (in mM): NaCl (105), $NH_4Cl$ (30), KCl (4.5), $CaCl_2$ (2), $MgCl_2$ (1), HEPES (20), glucose (11), pH adjusted to 7.4 with 4 M NaOH at 37°C. After 6 min, the superfusate was returned to normal Tyrode. In some experiments, the solutions contained 30 μM cariporide to block NHE1 (Tocris, Cat# 5358), as control. In separate experiments, cSNARF1 fluorescence ratio was calibrated using the nigericin method and this curve was used to convert measured cSNARF1 ratio to pHi.

## Measuring resting pHi

Cells were seeded at a density of 80,000 cells/well in ibidi flat-bottom 96-well plates to produce a monolayer after 24 hr. On the measurement day, cells were loaded dually with cSNARF1 (5 µg/mL) and Hoechst 33342 (1:1000) for 15 min, followed by replacement with bicarbonate-free, Phenol Red-free medium based on D5030, containing 10 mM HEPES and MES and titrated at pH over the range 6.2–7.7. Plates were imaged at Cytation 5 plate reader and each cell, identified from the location of its nucleus, was analyzed for pHi (*Michl et al., 2019*). To measure variation of pHi, the frequency distribution was offset to the mode pHi for each experimental condition.

## Measuring medium acidification

Cells were seeded at densities of 2000 cells/well on clear, flat-bottom 96-well plate (ibidi). Cells were cultured in bicarbonate-buffered media (D7777, Sigma) set to a pH of 7.4 by adjusting $NaHCO_3$ accordingly. The cells were incubated for 4 days at 37°C with 5% $CO_2$ before media were removed for glucose and lactate measurements.

## Pentra assay for glucose and lactate

Cells were seeded onto 96-well plates and grown in DMEM (D7777, Sigma) at pH 7.4. After 5 days, medium was collected, spun to remove residue, and analyzed in a Pentra C400 for glucose and lactate concentrations. Calibrations used standard solutions.

## Measuring metabolic fluxes

Two protocols were performed to measure metabolic rate. The first interrogated glycolytic flux, and the second interrogated glycolysis and respiration simultaneously. For the first, cells were cultured at high density (70,000 cells/well) in flat-bottom, black 96-well plates. To report extracellular pH, media contained 50 µM cSNARF1-dextran. Media, based on DMEM D5030, contained 25 mM glucose, 10% FBS, 1% PS, 1 mM pyruvate, 1% GlutaMAX, and 2 mM HEPES and 2 mM MES to provide a low but constant buffering power over the pH range studied. Fluorescence was monitored for 17 hr using a Cytation 5 device (BioTek, Agilent, Winooski, VT). Excitation was provided by a monochromator, and fluorescence emission was detected sequentially at wavelengths optimized as described previously (*Blaszczak et al., 2021*). For the second method, cells were cultured at high density (70,000 cells/well) in flat-bottom, black 96-well plates. To report extracellular pH and $O_2$, media contained 2 µM HPTS (8-hydroxypyrene-1,3,6-trisulfonic acid trisodium salt) and 50 µM RuBPY (tris(bipyridine)ruthenium(II) chloride). Media, based on DMEM D5030, contained 25 mM glucose, 10% FBS, 1% PS, 1 mM pyruvate, 1% GlutaMAX, and 2 mM HEPES and 2 mM MES to provide a low but constant buffering power over the pH range studied. Prior to measurements, each well was sealed with 150 µM mineral oil to restrict $O_2$ ingress. HPTS and RuBPY fluorescence were monitored for 17 hr using a Cytation 5 device (BioTek, Agilent). Excitation was provided by a monochromator, and fluorescence emission was detected sequentially at five wavelengths, which were optimized as described previously (*Blaszczak et al., 2021*).

## SRB and GFP growth assays

GFP-labeled cells (pLV-eGFP addgene plasmid# 36083) were transduced with lentiviral constructs. Then, 48 hr later, cells were seeded at 2000 cells/well on flat-bottom 96-well plates (ibidi). For co-cultures, the ratio of the two cellular populations was varied. The following day, medium was replaced with bicarbonate-buffered media (D7777, Sigma) and incubated at 37°C with 5% $CO_2$ for 7 days; in some experiments, atmospheric $O_2$ was reduced to 2% to produce hypoxia. For some experiments, incubations were terminated on days 2–7 to obtain a time course. Plates were analyzed for GFP fluorescence and SRB absorbance (Cytation 5). The GFP signal was measured in intact monolayers (excitation 490 nm, emission 520 nm). Next, wells were prepared for the SRB assay by fixing with PFA 4% for 30 min, washing with water and staining with 100 µL 0.057% SRB (in 1% acetic acid) for 30 min. Residual SRB was removed by washing with 1% acetic acid four times. Finally, 200 µL/well 10 mM Tris base was added to release SRB and measure absorbance (520 nm).

## In vivo xenograft experiments

Six-week-old female athymic Nude Crl:NU(NCr)-Foxn1nu mice were injected subcutaneously with a 1:1 mixture of either WT DLD1 cells and GFP-labeled *NDUFS1* KO SW1222 cells or a 1:1 mixture of

*GJB2* KO DLD1 cells and GFP-labeled *NDUFS1* KO SW1222 cells. CRCs cells were resuspended in 100 µL of a 1:1 mixture of Matrigel and serum-free DMEM medium before injection. Each mouse was injected with 2 million cells to each flank: WT DLD1 + *NDUFS1* KO SW1222 on left and *GJB2* KO DLD1 cells + *NDUFS1* KO SW1222 on right. Mice were weighed and tumors were measured three times a week. At the end of the experiments, when tumors reached the scientific or humane endpoint, mice were sacrificed and tumors excised and processed for imaging. Tumor cryo-sections were cut (Thermo Fisher Scientific cryostat) across the tumor height to slices of thickness 5 microns. Sections were then stained with Hoechst and anti-GFP polyclonal antibody (Thermo Fisher #A-11122) to enhance the signal from GFP-expressing *NDUFS1*-deficient SW1222 cells. Briefly, frozen slides were air-dried for 30 min, relevant sections marked with a PAP pen ring, and then slides were dip-washed in PBS for 10 min (staining jar), dried with tissue and placed in staining chamber tray, blocked with PBS Triton 0.1% with 5% goat serum and 2% BSA for 30 min, rinsed with PBS 3× (using 3 mL pipette), incubated with primary anti-GFP antibody in PBS Triton 0.1% at 1:250 dilution and 2% goat serum for 1 hr, rinsed 3× with PBS, incubated with secondary antibody (goat anti-rabbit Alexa 488) at 1:1000 and Hoechst-33258 in PBS Triton 0.1% for 30 min, rinsed 1× with PBS and then dip-washed in PBS for 2 min, then dried with tissue. Approximately 35 µL of slow-fade diamond (Thermo Fisher, S36963) was added and a coverslip applied and sealed with nail polish. Every fifth slides was imagined using a Zeiss Axioscan slide scanner using a ×20 0.8 Plan-Apochromat objective. The montage of images was constructed for the Hoechst and GFP channels. To demarcate the area of the tumor, the Hoechst signal was dilated (rolling ball, 50 pixels) to expand beyond nuclei and fused to form a mask, with smaller particles ignored. The background of GFP images was calculated from the mean of fluorescence within the Hoechst-defined tumor mask. To produce a thresholded GFP image, pixels were defined as GFP-positive if their fluorescence exceeded 4, 5, or 6 standard deviations from the mean GFP signal within the masked region; all other pixels were reassigned a GFP value of zero. Next, all GFP-positive clusters were counted (ignoring large particles that are likely noncellular stains) and the total area calculated. Each slide also provided a measure of tumor area. The total GFP-positive cluster count and area, and overall tumor area were summed over all slices. The analyses were repeated for the three levels of thresholding to ensure the findings are consistent.

## Statistics

Data are expressed as mean ± SEM. Significance *p<0.05, **p<0.01, ***p<0.001. GMM was performed using GMMchi (https://github.com/jeffliu6068/GMMchi; *Liu, 2022*). Before selecting tests, data were tested for normality by the Kolmogorov–Smirnov test. One-sample *t*-tests were performed to compare data against an expected value (for data failing normality test, the nonparametric test was the Wilcoxon signed-rank test). For comparisons between two or more samples, one-way ANOVA was performed.

## Acknowledgements

In memory of Robert J Gillies (1953-2022) who made milestone contributions to the field of cancer physiology and would have enjoyed discussing the present work with us. The work was supported by the European Research Council, SURVIVE #723997. We thank Professor Alison Simmons and Dr. Agne Antanaviciute for access to the single-cell RNA GSE datasets for colonic epithelium.

## Additional information

### Funding

| Funder | Grant reference number | Author |
| --- | --- | --- |
| European Research Council | 723997 | Pawel Swietach |

The funders had no role in study design, data collection and interpretation, or the decision to submit the work for publication.

## Author contributions
Stefania Monterisi, Data curation, Formal analysis, Investigation, Methodology, Writing – review and editing; Johanna Michl, Resources, Formal analysis, Investigation, Methodology, Writing – review and editing; Alzbeta Hulikova, Resources, Formal analysis, Writing – review and editing; Jana Koth, Esther M Bridges, Resources, Data curation, Formal analysis, Methodology; Amaryllis E Hill, Formal analysis, Investigation, Methodology; Gulnar Abdullayeva, Formal analysis; Walter F Bodmer, Conceptualization, Resources, Formal analysis, Writing – review and editing; Pawel Swietach, Conceptualization, Resources, Data curation, Software, Formal analysis, Supervision, Funding acquisition, Validation, Investigation, Visualization, Methodology, Writing - original draft, Project administration

## Author ORCIDs
Stefania Monterisi http://orcid.org/0000-0002-8802-9742
Gulnar Abdullayeva http://orcid.org/0000-0002-4961-3671
Walter F Bodmer http://orcid.org/0000-0001-6244-9792
Pawel Swietach http://orcid.org/0000-0002-9945-9473

## Ethics
This study was performed in accordance with the provisions of the Animals (Scientific Procedures) Act 1986 and recommendations set by the Biomedical Services unit at Oxford University. All work involving mice obtained approval of ethics and welfare board instructions, and was authorized by Project Licence PPL P01A04016 issued by the UK Home Office. All surgery was performed under anesthesia, and appropriate post-recovery care was provided to minimize suffering.

## Decision letter and Author response
Decision letter https://doi.org/10.7554/eLife.78425.sa1
Author response https://doi.org/10.7554/eLife.78425.sa2

# Additional files

## Supplementary files
• Supplementary file 1. Correlation between connexin expression and functional coupling.
• Transparent reporting form
• Source data 1. Contains full length scans of western blots included in this publication.

## Data availability
Figure 1 - original sequencing data is available from the reference given in text. A tabulated form of Figure 1E is given as a supplementary file. All original data generated or analysed during this study are included in the manuscript and supporting file; source data files have been provided.

The following previously published dataset was used:

| Author(s) | Year | Dataset title | Dataset URL | Database and Identifier |
|---|---|---|---|---|
| Fawkner-Corbett D, Antanaviciute A, Parikh K, Jagielowicz M, Geros AS, Gupta T | 2021 | Spatiotemporal analysis of human intestinal development at single-cell resolution | https://www.ncbi.nlm.nih.gov/geo/query/acc.cgi?acc=GSE116222 | NCBI Gene Expression Omnibus, GSE116222 |

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

## Appendix 1

### Calculating permeability from FRAP recordings

Permeability to calcein across cell-to-cell junctions was measured in monolayers where cells made contacts between one another. A cell within a confluent structure was selected for bleaching. Permeability to calcein, in units of μm/s, was measured from the FRAP recovery time course measured in the bleached cell, and geometrical information obtained from the image of the monolayer.

The principle behind calculating permeability P is *equation 1*:

$$dC_{cell}/dt = \rho \cdot P \cdot (C_{cell} - C_{surround}) \tag{1}$$

where subscripts 'cell' and 'surround' refer to the central cell and its cellular neighbors, respectively, and $\rho$ is the ratio of the surface area of the barrier (i.e., membrane) and the cell volume.

In the experiment, fluorescence (F) is a readout of concentration (C), normalized by a scalar α:

$$C = \alpha \cdot F$$

Thus, the rate of recovery (LHS of *Equation 1*) is

$$dC/dt = d(\alpha \cdot F)/dt = \alpha \cdot dF/dt \tag{2}$$

And the RHS of *Equation 1* becomes

$$P \cdot (C_{cell} - C_{surround}) = P \cdot (\alpha \cdot F_{cell} - \alpha \cdot F_{surround}) = \alpha \cdot P \cdot (F_{cell} - F_{surround}) \tag{3}$$

Putting 2 and 3 together cancels out α:

$$dF_{cell}/dt = \rho \cdot P \cdot (F_{cell} - F_{surround})$$

Prior to bleaching, there are no net gradients, thus initial $F_{cell}$ and $F_{surround}$ are equal. To account for this, fluorescence F in the cell and its surroundings can be normalized to starting signals:

Thus, *Equation 1* becomes

$$df_{cell}/dt = \rho \cdot P \cdot (f_{cell} - f_{surround})$$

P can therefore be expressed as:

where $df_{cell}/dt$ is measured from the fluorescence recovery time course. This is the initial rate of fluorescence calculated by best-fitting the recovery time to a monoexponential curve and calculating the initial gradient at the moment of bleaching. Gradient $f_{cell}$-$f_{surround}$ is measured experimentally; in fact, it is set by the protocol to be numerically 0.5 because the central cell is bleached until fluorescence decreases to 50%.

An accurate calculation of $\rho$ would require volume rendering. However, it can be approximated by a reasonable assumption that the monolayer is cuboidal and contracts are made at the sides of cells (and not above or below the x–y plane). The monolayer image provides information about the area of the cell in the x–y plane, and its perimeter then makes contact with neighboring cells. The cell's volume is thus area × height, and the cell's surface (at which its contacts are made) is the cell's perimeter × height. Thus, for the bleached cell,

$$\rho = \text{perimeter} \times \text{height} / \text{area} \times \text{height} = \text{perimeter/area}.$$

All these values are inferred from the data. The permeability refers specifically to calcein.

### Calculating coupling coefficient from CellTracker co-cultures

Monolayers are imaged sequentially to obtain Violet and Orange fluorescence bitmaps (512 × 512 pixels) at the optimal wavelength settings for CellTracker Violet and Orange, respectively. Each experiment consisted of three conditions: (i) cells that were loaded with Violet only, (ii) cells that were loaded with Orange only, and (iii) a co-culture of cells that had been loaded with Violet or Orange, mixed in 1:1 ratio. Thus, (i) and (ii) are a mono-culture, whereas (ii) is expected to be a well-mixed co-culture. Diffusive exchange across gap junctions is expected to lead to mixing of dyes between cells. This will have no effect on the mono-cultures as like is exchanged for like, but in co-cultures, some cells would contain Violet and Orange pixels. The degree of dye mixing was calculated by relating

the Violet and Orange signal on a pixel-by-pixel basis. To ensure accuracy, several image processing steps are necessary.

*Violet mono-culture image processing*:

1. In the Violet fluorescence image:
   a. Background pixels are identified by thresholding. This is used to estimate confluency and the inverse to identify cells.
   b. The signal in nonbackground pixels is averaged to give a mean Violet fluorescence value in cells. This is then used for normalizing the degree of Violet dye loading, which may be different to the Orange signal. This is called $F_V^{mono}$.
2. In the paired Orange fluorescence image, the 95% percentile of signal in nonbackground pixels is calculated to describe background Orange fluorescence. This represents autofluorescence in the Orange range in the absence of CellTracker Orange. This value is called $B_O^{mono}$.
3. The process is repeated for all images taken in the same back (biological replicate) to obtain a mean $F_V^{mono}$ and $B_O^{mono}$.

*Orange mono-culture image processing*:

1. In the Orange fluorescence image:
   a. Background pixels are identified by thresholding. This is used to estimate confluency and the inverse to identify cells.
   b. The signal in nonbackground pixels is averaged to give a mean Orange fluorescence value in cells. This is then used for normalizing the degree of Orange dye loading, which may be different to the Violet signal. This is called $F_O^{mono}$.
2. In the paired Violet fluorescence image, the 95% percentile of signal in nonbackground pixels is calculated to describe background Violet fluorescence. This represents autofluorescence in the Violet range in the absence of CellTracker Orange. This value is called $B_V^{mono}$. Typically, $B_V^{mono}$ is larger than $B_O^{mono}$.
3. The process is repeated for all images taken in the same back (biological replicate) to obtain a mean $F_O^{mono}$ and $B_V^{mono}$.

Once complete, means $F_V^{mono}$ is background-subtracted by $B_V^{mono}$, and similarly for Orange.

*Co-culture image processing*:

1. Background pixels are identified as those that are below the threshold for Violet and Orange fluorescence.
2. The nonbackground pixels are taken as areas within cells. Each pixel is annotated with a Violet and Orange fluorescence. These signals are background-subtracted ($B_V^{mono}$ and $B_O^{mono}$, respectively), and then normalized to mean signals ($F_V^{mono}$ and $F_O^{mono}$, respectively). Each pixel is now described by a background-subtracted and normalized fluorescence array, $F_V$ and $F_O$.
3. To quantify exchange, Violet fluorescence is expressed as a fraction of total fluorescence ($F_V + F_O$) and, similarly, Orange fluorescence is expressed as a fraction of total fluorescence ($F_V + F_O$). The smaller of the two values is taken to represent coupling coefficient, as the smaller value relates to fluorescence from the ingress of CellTracker dye from neighboring cells, rather than dye that had been loaded originally in the cell. To scale this coefficient on a scale from 0 to 1, the final value is doubled.

