## [Editor Report]

This innovative study is of potential interest to a broad readership of cancer biology by addressing an important mechanism regarding how spontaneous mutations in cancer cells affecting metabolic pathways do not necessarily result in a functional defect thanks to gap junctionally-mediated exchange of metabolites.

---

## [Decision Letter]

**Decision letter after peer review:**

Thank you for submitting your article "Solute exchange through gap junctions lessens the adverse effects of inactivating mutations in metabolite-handling genes" for consideration by *eLife*. Your article has been reviewed by 3 peer reviewers, one of whom is a member of our Board of Reviewing Editors, and the evaluation has been overseen by Aleksandra Walczak as the Senior Editor. The following individual involved in review of your submission has agreed to reveal their identity: Bruce J Nicholson (Reviewer #2).

Essential revisions:

Overall, the study was found to be important, and well designed and executed. A related question to address is if Cx31 and 43, which do not seem to have the same roles in maintaining cell coupling as Cx26, also play any role in such metabolic rescue. Additional experiments are needed to address the Cx31 and 43 KD effects, particularly western blots on these two Cxs to prove if adding both shRNAs to Cx43 and 31 also results in reduced expression (or for that matter in KD of any of the three Cxs affects any of the others). Some of gap junctional coupling measurements are qualitative, rather than quantitative, and should be presented as such. Additionally, metabolic coupling and by-stander effect, similar to what is seen here, have been extensively reported; these important contributions ought to be thoroughly discussed including citations of important literature that are not currently included. A more, clear elaboration is needed to explain the novelty of the study and how these in vitro findings recapitulate the types of actual somatic mutations in cancer growth under clinical scenarios.

*Reviewer #1 (Recommendations for the authors):*

On top of page 9, it is not clear if targeting a single connexin gene like GJA1 as stated would be a more viable approach since among the expression of other connexin isoforms, Cx26 is a dominant connexin that forms gap junctions in most of the CRC cells.

*Reviewer #2 (Recommendations for the authors):*

The conclusions from this work are of strong interest to the field, but the authors should do a better job of pointing out how this is really the logical opposite effect of the bystander effect. Which has been demonstrated to be dependent on gap junctions for many years. This does not, however, detract from the current work, but is important for placing it in context.

The work is quite extensive and represents a very cleverly constructed to study such a complex subject. Complex results are generally well presented, with the exception of the pH effects presented in Figures5 G-H, which are not easy to understand. Also, the terms used to describe permeability are misleading and inappropriate. Permeability has a specific meaning, and requires knowledge of the concentration gradient, the relative volumes of the cells and control over the number and extent of interfaces across which permeation occurs – none of which apply in this case. What is actually measured is a rate of FRAP, which is loosely correlated with permeability of gap junctions. With regard to coupling coefficient, this has a specific meaning in the literature that is usually related to simple transfers between paired cells. The meaning of the term here should be better explained. In addition, it appears that some cells do not partake in the dye mixing, and it may be of use to know if these represent a subset of cells that do not express connexins, which could be established through IF labeling after the transfer.

In Figure 2 it is clearly shown that KD of Cx26 impairs coupling, but KD of Cx31 or 43 do not (and may even increase coupling). Yet this is then ignored and focus shifts exclusively to Cx26. Given that these Cx31, and at least in a subset of cells Cx43, are expressed at similar levels to Cx26, this should be explored, as in terms of physiological relevance to the tissue this could be of great importance. Initial understanding of this result would require information on the efficacy of KD for Cx32 and 43, yet the authors only show results for Cx26 KD (Figure 2 H and J). It could also be useful to include at least one test of Cx31 KO in one of the metabolite systems studied subsequently in the paper. The others also need to be shown. At the very least, some discussion of this anomalous result should be presented. Points that could be considered are: Cx26 and 31 oligomerize, so perhaps the oligomer has different permeation than Cx26 monomers; This is not true for Cx43, but it appears that it is not expressed in all cells.

*Reviewer #3 (Recommendations for the authors):*

Specific comments

1. In the abstract the authors' write:

Line 35 – Function rescue was dependent on Cx26 channels and reduced phenotypic heterogeneity among cells.

In isolation this sentence seems to be a strong statement. The authors probably did not mean it to come across this way and likely view that it should be interpreted in the context of prior sentences. However, as a standalone inference it should be toned down – and also other places in the manuscript where related wording occurs. The authors only looked carefully at Cx26, excluding other connexins. It is not clear that other connexins can not also contribute to this phenomenon, or that important nuances may occur in their models with heteromeric and/or heterotypic patterns of connexin contact.

2. In a related vein, that another Cx was examined (e.g., Cx43) appears to be a gap in generalizing the results from the study.

3. To obtain Cx loss of function the authors rely almost exclusively on knockdown (KD). This is OK – but it does raise the prospect of phenotypic changes occurring over the multi-hour time course to effective KD/loss of function. There are peptides that enable relatively selective and rapid (within minutes) loss or gain of function of Cx function. Also, effective and quick, though somewhat less selective, are pharmacological agents that can block or enhance GJ junction. Why are such approaches not employed to address this issue?

4. The explanation of the statistical approach was minimal. More details are required on the nature of the analyses undertaken. Normal, non-parametric? If normal, was testing for homogeneity of variance or normality performed? Was post-hoc testing performed? A review by a professional statistician may be required.

5. The authors' write:

Line 321- Intriguingly, the exact magnitude of permeability did not predict the degree to which solutes can be equilibrated between cells at steady-state. This is because even weakly coupled confluent monolayers were able to exchange dyes, if allowed sufficient time. Thus, the phenomenon of solute equilibration across coupled cancer cells is likely to be significant even in cells with low connexin expression.

Some care here needs to be taken in this reviewer's opinion. The authors model involves intermingled cultures in which diffusion typically occurs for distances of not more than a few cell diameters. The situation may be different in cancer in vivo (e.g., in solid tumors) or 3D cultures – where diffusion distance from competent non-cancer cells will likely present a more exacting path for a diffusing metabolite. Have the authors considered an approach for replicating the situation of action at distance in vitro in their model? Without such data the statement "…the phenomenon of solute equilibration across coupled cancer cells is likely to be significant even in cells with low connexin expression" Is really only supported by the data in hand within the tight confines of the models used in these studies.

6. A diagram summarizing findings may help readers efficiently determine what is new and significant about this study -information that may help address the opening point on differentiation from the already longstanding concept of "metabolic cooperation".

[Editors’ note: further revisions were suggested prior to acceptance, as described below.]

Thank you for resubmitting your work entitled "Solute exchange through gap junctions lessens the adverse effects of inactivating mutations in metabolite-handling genes" for further consideration by *eLife*. Your revised article has been evaluated by Aleksandra Walczak (Senior Editor) and a Reviewing Editor.

The reviewers appreciate your detailed responses to the previous review, revised text and additional experiments. The manuscript has been improved. Please address two remaining issues raised by Reviewer #2.

*Reviewer #2 (Recommendations for the authors):*

The authors have done an excellent job in responding to reviewer's suggestions, even including additional experiments that directly address some of the issues, as well as appropriately modifying the text. As noted before, this is an important extension of prior work that establishes a novel explanation as to why defects in essential genes that produce gap junctionally permeant metabolites are not selected against in tumor populations (or presumably any well coupled cell populations). As such, it is highly appropriate for publication in *eLife*.

Only two issues remain that should be addressed:

1. The authors persist in using the term "Permeability", measured in units of uM/min for their FRAP data. the explanation of its quantitation is now significantly extended in Appendix 1, and this is a valuable addition. However, as is evident in the calculation, The α factor in equations 2 and 3 converting fluorescence to concentration cancels in calculating the final permeability (rho), so there is no concentration information in rho. So while the permeability numbers are good for comparative purposes, as used here, it is inappropriate to list permeability as uM/min, as nowhere is the concentration of Calcein inside the monolayer established. In addition, the comparison BETWEEN cell lines would require that the initial concentration of calcein in the monolayer be independently established for each cell line (as the uptake of calcein in each cell line is likely different and unlikely to reach equilibrium in the 10 min incubation used). Otherwise, a caveat would need to be included in such inter-cell type comparisons. All of this being said, the assay is nicely described, and still useful for comparisons as used here. It just does not represent an actual permeability in terms of uM/min.

2. The authors have done a creditable effort to demonstrate that Cx31 appears not to be anywhere near as effective in rescuing metabolite deficient phenotypes as Cx26. They also make the same claim for Cx43, but this is not supported by the data they present. They test if ALDO KO DLD1 (Figure 6) or NDUFS1-KO SW1222 (Figure 7) can be rescued by wt cells expressing Cx43 (C10), stating that "These co-cultures should, in theory, produce heterotypic connexins (should read gap junctions)". The problem is they should and likely do NOT, as Cx26 and 43 have been demonstrated to NOT form heterotypic channels, and the authors demonstrate earlier that neither DLD1 or SW1222 express Cx43. To demonstrate that Cx43 can or cannot mediate the metabolic rescue they are studying, they would need to do the Aldolase of NDUFS1 KOs in C10 cells, and then do cultures of the KO and wt cells, all of which express Cx43. Otherwise they need to limit their conclusions about lack of metabolic rescue to Cx31, which does form heteroptypic channels with Cx26. This is important, as Cx43 is also commonly expressed in tumors.

---

## [Author Response]

Essential revisions:Overall, the study was found to be important, and well designed and executed. A related question to address is if Cx31 and 43, which do not seem to have the same roles in maintaining cell coupling as Cx26, also play any role in such metabolic rescue.

To test for metabolic rescue by Cx isoforms other than Cx26, we now provide additional experiments using CRC cells that express Cx43 or Cx31, but have low or absent Cx26. Cells meeting this criterion included C10 and NCIH747, respectively (Figure 1E). We performed two types of experiments: In the first, we co-cultured *ALDOA*-deficient cells with their wild-type counterparts (Figure 6H). For the second experiment, we co-cultured *NDUFS1*-knockout SW1222 cells with either C10 or NCIH747 cells (Figure 7H). As an additional negative control, we provide new data using connexin-negative RKO cells. Overall, the results confirm that metabolic rescue is best achieved with Cx26 expressing cells. The lower effectiveness of Cx31 is consistent with our observations that *(i) GJB3* expression levels do not correlate with overall connectivity, and *(ii) GJB3* knockdown does not ablate coupling functionally, i.e. Cx31 channels are unlikely to form strong conductance pathways for metabolites between cells. In the case of Cx43, we find that monolayers grown from *GJA1*-expressing lines contained both Cx43-positive and negative cells (confirmed by IF, and now by FACS – new data in Figure 2H). Thus, not all cells of nominally *GJA1*-high cell lines will benefit from Cx43-dependent connectivity, and hence metabolic rescue. Thus, our findings strengthen the case for Cx26 being particularly effective in metabolic rescue, at least in CRC cells.

Additional experiments are needed to address the Cx31 and 43 KD effects, particularly western blots on these two Cxs to prove if adding both shRNAs to Cx43 and 31 also results in reduced expression (or for that matter in KD of any of the three Cxs affects any of the others).

This point has arisen from data shown in Figure 2D results. There, we show that coupling in selected cells is reduced by *GJB2* knockdown, but not *GJB3* knockdown. The rationale for this experiment was that expression of these two genes is often concordant, so a KD experiment was necessary to verify the functionally dominant isoform. There was a trend that *GJB3* knockdown increased coupling, but we explain that this was not statistically significant (*GJB3* v scrambled) when analysed by hierarchical (nested) methods (indeed, we did not indicate this to be significant in the original revision, and therefore did not study it further). The trend may result from a compensatory increase in other connexin isoforms but Cx31 knockdown did not appear to substantially change Cx26 or Cx43 levels (new blots in Figure 2-supp1). Moreover, *GJB3* (Cx31) knockdown did not induce *GJA1* (Cx43) expression in an otherwise negative line (DLD1). Similarly, Cx43 knockdown did not affect Cx31 or Cx26 levels (Figure 2-supp1). Moreover, in DLD1 Cx26 stable knockout cells, we find no major effect on Cx31 or Cx43 expression, relative to wild-type cells (Figure 2-supp1). It is possible that changes to connexin expression could alter the density of connections or confluency of cells, thereby strengthening Cx26-dependent connections, but this cannot be resolved with fluorescence imaging alone. Overall, we do not see compelling evidence for compensatory changes in connexin levels upon knockdown. It is plausible that *GJB3* knockdown reduces the incidence of Cx26/Cx31 heterotypic channels, which have been shown to be less permeable than Cx26 channels. Genetic ablation of Cx31 would therefore increase overall permeability. Given that Cx26 channels are the dominant source of metabolic rescue, we felt that studying gene regulation of other connexins would fall outside the scope of this study; we hope the reviewers agree.

Some of gap junctional coupling measurements are qualitative, rather than quantitative, and should be presented as such.

We provide an explanation of our method and explain how it can provide a reasonably quantitative estimate of ensemble permeability. This is because we combine geometrical data with flux data, and we expand on this further in the Methods. We agree that the original description did not provide sufficient information for the reviewers to judge the method’s robustness.

Additionally, metabolic coupling and by-stander effect, similar to what is seen here, have been extensively reported; these important contributions ought to be thoroughly discussed including citations of important literature that are not currently included.

We have revised the Introduction and Discussion to explain how our findings go beyond what is known about the bystander effect. We added key references for the discovery of the bystander effect and metabolic cooperation, and put these in the context of our tested hypothesis. Thank you for these valuable suggestions.

A more, clear elaboration is needed to explain the novelty of the study and how these in vitro findings recapitulate the types of actual somatic mutations in cancer growth under clinical scenarios.

We have expanded on the significance of our findings for cancer – elaborating further on what we had written about negative selection (specifically, how our data can explain the scarcity of negative selection in vivo, a hitherto unexplained finding). In recognition that our experiments were in vitro, we now add findings from animal xenografts that support our hypothesis.

Reviewer #1 (Recommendations for the authors):On top of page 9, it is not clear if targeting a single connexin gene like GJA1 as stated would be a more viable approach since among the expression of other connexin isoforms, Cx26 is a dominant connexin that forms gap junctions in most of the CRC cells.

We apologise for the poor wording. Our intention was to select cells with a range of Cx26 expression, without other major connexins. We now added data from other cell lines, thus this sentence is no longer accurate, and we have revised it.

Reviewer #2 (Recommendations for the authors):The conclusions from this work are of strong interest to the field, but the authors should do a better job of pointing out how this is really the logical opposite effect of the bystander effect. Which has been demonstrated to be dependent on gap junctions for many years. This does not, however, detract from the current work, but is important for placing it in context.

We have revised the text to refer to the by-stander effect, and indicate how the present work is distinct, and its logical extension.

The work is quite extensive and represents a very cleverly constructed to study such a complex subject. Complex results are generally well presented, with the exception of the pH effects presented in Figures5 G-H, which are not easy to understand.

This has now been addressed (see comments under weaknesses).

Also, the terms used to describe permeability are misleading and inappropriate. Permeability has a specific meaning, and requires knowledge of the concentration gradient, the relative volumes of the cells and control over the number and extent of interfaces across which permeation occurs – none of which apply in this case. What is actually measured is a rate of FRAP, which is loosely correlated with permeability of gap junctions.

We elaborate on this calculation: please see comments under weaknesses

With regard to coupling coefficient, this has a specific meaning in the literature that is usually related to simple transfers between paired cells. The meaning of the term here should be better explained. In addition, it appears that some cells do not partake in the dye mixing, and it may be of use to know if these represent a subset of cells that do not express connexins, which could be established through IF labeling after the transfer.

We explain the calculation of coupling coefficient and provide a definition (please see comments under weaknesses). The idea of performing IF is a good one; however, fixing cells will also wash away CellTracker, so the suggested analysis would not be feasible.

In Figure 2 it is clearly shown that KD of Cx26 impairs coupling, but KD of Cx31 or 43 do not (and may even increase coupling). Yet this is then ignored and focus shifts exclusively to Cx26. Given that these Cx31, and at least in a subset of cells Cx43, are expressed at similar levels to Cx26, this should be explored, as in terms of physiological relevance to the tissue this could be of great importance. Initial understanding of this result would require information on the efficacy of KD for Cx32 and 43, yet the authors only show results for Cx26 KD (Figure 2 H and J).

We apologize for not providing these blots in the first instance. Blots of lysates matching functional data are shown in Figure 2-suppl1+2 and confirm efficacy of Cx43 and Cx31 knockdown.

It could also be useful to include at least one test of Cx31 KO in one of the metabolite systems studied subsequently in the paper. The others also need to be shown. At the very least, some discussion of this anomalous result should be presented. Points that could be considered are: Cx26 and 31 oligomerize, so perhaps the oligomer has different permeation than Cx26 monomers; This is not true for Cx43, but it appears that it is not expressed in all cells.

We provide new data using cells that are Cx26-low/negative but express other connexins (added to Figure 6 and Figure 7). There, we show that connexins other than Cx26 are not as effective as Cx26 in enabling metabolic rescue. Specifically, we performed co-cultures of *NDUFS1*-KO SW1222 (Cx26+) cells with either C10 and NCIH747, i.e. Cx26-low lines. These co-cultures should, in theory, produce heterotypic connexins but there was no significant rescue of *NDUFS1*-deficient cells. These new data demonstrate the high effectiveness of Cx26 channels in enabling metabolic rescue. Strikingly, a recent analysis by Robert Gatenby and colleagues showed that only Cx26 is ‘never mutated in cancers’, i.e. is absolutely necessary for cancers to thrive. This is also the connexin isoform that is native to the parent epithelium, and thus may be most suited to sustain communication, even after mutations.

The issue of oligomerisation between Cx isoforms is interesting, and thank you for raising this point. Indeed, it is plausible that weakly-conducting Cx26/Cx31 channels are formed in CRC, and the apparent strengthening of coupling in Cx31 KD cells could be explained as a shift away from heterotypic channels, towards higher conductance homotypic Cx26 channels. However, as explained above, hierarchical analysis does not detect a statistically significant effect between scrambled controls and *GJB3* knockdown cells, so the issue is unresolved. Nonetheless, the firm conclusion of the work is that *GJB2* KD (as well as KO for DLD1) strongly uncouples monolayers.

Reviewer #3 (Recommendations for the authors):Specific comments1. In the abstract the authors' write:Line 35 – Function rescue was dependent on Cx26 channels and reduced phenotypic heterogeneity among cells.In isolation this sentence seems to be a strong statement. The authors probably did not mean it to come across this way and likely view that it should be interpreted in the context of prior sentences. However, as a standalone inference it should be toned down – and also other places in the manuscript where related wording occurs. The authors only looked carefully at Cx26, excluding other connexins. It is not clear that other connexins can not also contribute to this phenomenon, or that important nuances may occur in their models with heteromeric and/or heterotypic patterns of connexin contact.

Thank you for these suggestions. We have toned down the abstract, and reviewed the text for inaccurate wording. We have added new data showing that Cx26-low/negative cell lines expressing other connexins (e.g. Cx43-positive C10 or Cx31/Cx45 positive NCIH747) cannot produce the same degree of rescue as Cx26 channels in DLD1 or similar CRCs. These new data are added to Figure 6/7. Further, we provide new data (Figure 7H) showing that heterotypic connections (e.g. between SW1222 and C10 or SW1222 and NCIH747) do not have the same efficacy in metabolic rescue as homotypic Cx26 connections (e.g. in SW1222 mono-cultures). Cx43 channels can produce strong connectivity in some cell lines (e.g. C10), but we observed considerable heterogeneity among cells in terms of their Cx43 levels (first inferred from IF images). We now confirm the presence of Cx43-low and Cx43-high sub-populations by FACS (new data; Figure 2H). Based on these findings, we argue that only some cells of a nominally *GJA1-*high cell line could potentially benefit from metabolic rescue. We believe that this pattern explains why Cx43 is less efficacious for metabolic rescue in a network of CRCs.

2. In a related vein, that another Cx was examined (e.g., Cx43) appears to be a gap in generalizing the results from the study.

To address this, we have added new data using C10 cells, a Cx26-low but strongly Cx43-positive line. Due to heterogeneity in Cx43 levels among cells, this line shows considerable variation in functional coupling as measured by FRAP (Figure 2). This cell was unable to rescue genetic deficiencies in *ALDOA* (homotypic Cx43; Figure 6) or *NDUFS1* (potentially heterotypic Cx26-Cx43; Figure 7). The original IF images in Figure 2G suggested that Cx43 expression is heterogenous among cells, with only a subpopulation being clearly positive. This is in contrast to the more uniform abundance of Cx26. We speculate that non-uniform Cx43 expression in a monolayer may limit the ability of this connexin to rescue metabolism globally, as it would require two random events to coincide: a somatic loss-of-function mutation *and* expression of Cx43. For this reason, Cx43 may be less effective in metabolic rescue than Cx26. To confirm this heterogeneity, we sorted *GJA1*-high LOVO and Caco2 cells by Cx43 levels, and found distinct sub-populations of dramatically different Cx43 levels, even after allowing cells to expand for western blotting. We add these new data to Figure 2. Note, however, that after prolonger culture and multiple passages, the cells return to the original, heterogenous population (thus it is not possible to keep a purely positive or negative population for experiments, such as rescue, over several days).

3. To obtain Cx loss of function the authors rely almost exclusively on knockdown (KD). This is OK – but it does raise the prospect of phenotypic changes occurring over the multi-hour time course to effective KD/loss of function. There are peptides that enable relatively selective and rapid (within minutes) loss or gain of function of Cx function. Also, effective and quick, though somewhat less selective, are pharmacological agents that can block or enhance GJ junction. Why are such approaches not employed to address this issue?

We concur that peptides and drugs are available for blocking Cxs. However, cancer-relevant readouts such as cell growth take days to develop, and therefore a genetic alteration is preferred over pharmacology. Moreover, some Cx inhibitors (e.g. carbenoxolone) require relatively high concentrations for efficacy, which raises concerns about off-target effects. Some blockers (e.g. glycyrrhetinic acid) also affect enzymes involved in signalling, which may impact cancer growth.

4. The explanation of the statistical approach was minimal. More details are required on the nature of the analyses undertaken. Normal, non-parametric? If normal, was testing for homogeneity of variance or normality performed? Was post-hoc testing performed? A review by a professional statistician may be required.

We have now expanded on the statistics. We performed normality tests, and elaborate on the methods used..

5. The authors' write:Line 321- Intriguingly, the exact magnitude of permeability did not predict the degree to which solutes can be equilibrated between cells at steady-state. This is because even weakly coupled confluent monolayers were able to exchange dyes, if allowed sufficient time. Thus, the phenomenon of solute equilibration across coupled cancer cells is likely to be significant even in cells with low connexin expression.Some care here needs to be taken in this reviewer's opinion. The authors model involves intermingled cultures in which diffusion typically occurs for distances of not more than a few cell diameters. The situation may be different in cancer in vivo (e.g., in solid tumors) or 3D cultures – where diffusion distance from competent non-cancer cells will likely present a more exacting path for a diffusing metabolite. Have the authors considered an approach for replicating the situation of action at distance in vitro in their model? Without such data the statement "…the phenomenon of solute equilibration across coupled cancer cells is likely to be significant even in cells with low connexin expression" Is really only supported by the data in hand within the tight confines of the models used in these studies.

This is a very good point. Our reasoning is that in a human cancer, mutations will be sporadic and therefore highly localized. Thus, the potential diffusion distance over which rescue would happen is small – perhaps a few cells, and not radically different to the in vitro situation. We have added this reasoning to the Discussion.

In light of this comment, we felt it was important to perform a series of xenograft experiments to confirm our in vitro findings in a more physiological scenario. The new in vivo data are shown in Figure 8. Xenografts were grown on the two flanks of mice and consisted of admixtures of respiring DLD1 cells and mitochondrially-defective (*NDUFS1* KO) SW1222 cells. We show that wild-type (Cx26-expressing) DLD1 cells can rescue the growth of *NDUFS1*-cells, which otherwise grow very poorly. However, a xenograft established using DLD1 *GJB2*-KO cells (instead of wild-type DLD1 cells) was unable to produce the same degree of rescue. This finding shows that rescue requires Cx26 channels between respiring DLD1 and mitochondrially-defective SW1222 cells.

6. A diagram summarizing findings may help readers efficiently determine what is new and significant about this study -information that may help address the opening point on differentiation from the already longstanding concept of "metabolic cooperation".

Thank you for this suggestion. We include a summary cartoon on the last figure (Figure 8), and expanded the Discussion accordingly.

[Editors’ note: further revisions were suggested prior to acceptance, as described below.]

Reviewer #2 (Recommendations for the authors):The authors have done an excellent job in responding to reviewer's suggestions, even including additional experiments that directly address some of the issues, as well as appropriately modifying the text. As noted before, this is an important extension of prior work that establishes a novel explanation as to why defects in essential genes that produce gap junctionally permeant metabolites are not selected against in tumor populations (or presumably any well coupled cell populations). As such, it is highly appropriate for publication in eLife.Only two issues remain that should be addressed:1. The authors persist in using the term "Permeability", measured in units of uM/min for their FRAP data. the explanation of its quantitation is now significantly extended in Appendix 1, and this is a valuable addition. However, as is evident in the calculation, The α factor in equations 2 and 3 converting fluorescence to concentration cancels in calculating the final permeability (rho), so there is no concentration information in rho. So while the permeability numbers are good for comparative purposes, as used here, it is inappropriate to list permeability as uM/min, as nowhere is the concentration of Calcein inside the monolayer established. In addition, the comparison BETWEEN cell lines would require that the initial concentration of calcein in the monolayer be independently established for each cell line (as the uptake of calcein in each cell line is likely different and unlikely to reach equilibrium in the 10 min incubation used). Otherwise, a caveat would need to be included in such inter-cell type comparisons. All of this being said, the assay is nicely described, and still useful for comparisons as used here. It just does not represent an actual permeability in terms of uM/min.

We appreciate the concerns raised, and kindly provide further explanations. We think there is a misunderstanding about the units. The Reviewer writes about “uM/min”, which is a unit of *flux* (concentration/time). Such a parameter would need accurate measurements of concentration. However, our reporting unit in Figure 2 is “um/min” (length/time), which is the standard unit of *permeability* (cf the standard unit of diffusion coefficient is um^2^/min, and can also be quantified by FRAP). This unit has:

i) a *geometrical aspect* that accounts for the extent of cell-cell contacts (unit: um), and

ii) a *temporal aspect* that is derived from the time-constant of fluorescence recovery (unit min^-1^)

Thus, our unit is independent of concentration. Appendix 1 clarifies why this is so.

We double checked our axis labels and confirm that we used micron per min for all the data.

We’d like to clarify some points made in the comment, please:

1) Correct: there is no concentration information in rho. Rho is purely geometrical and has no concentration data – it is calculated from cell outlines and indeed, could have been measured from brightfield images. Units: um^-1^. This accounts for the “geometric” component of P.

2) We used a proportional measure of intracellular calcein levels to calculate a recovery time constant after bleaching (the “time” component of P), in units of min^-1^. This time constant is, by definition, independent of concentration. The same recovery time constant would be obtained irrespective of whether we chose uM, mM or (as in our case) fluorescence units on the y-axis. Thus, a calibration from fluorescence units to (real) concentration is not required.

3) The initial concentration of calcein in μm (or similar) is not a factor in calculating permeability from FRAP. In Appendix 1, we show that concentration features on the left and right side of equation 1: thus, the unit of concentration is irrelevant, as long as it’s the same on both sides. Thus, we could assess calcein abundance in uM, mM or fluorescence units, and still arrive at the same answer. [NB: this assumes linearity between fluorescence and concentration – a reasonable assumption for dyes in sub-mM levels, like in cells]. Please consider a thought experiment: if we loaded cells with half the amount of calcein, and bleached half of this amount in the central cell, the recovery of fluorescence would be unaffected. Geometry will, of course, be the same, thus P will be unchanged.

Since we have used um/min in all our previous studies of gap junctions over the past two decades, we would like to retain this convention, please. Many authors have chosen to quantify FRAP in terms of a time constant only, but this ignores geometry, which will be different between cell lines – e.g. some cells form denser contacts. We believe our approach of combining recovery and geometry is more appropriate.

Multiple FRAP studies of cytoplasmic properties have converted recovery time constants into a diffusion coefficient (e.g. of fluorescent dyes), in units of um^2^/min. Diffusion, like permeability, is independent of initial concentration: at 100 mM or 10 mM, Na^+^ ions still have the same diffusion coefficient (at constant temperature).

We do, however, accept that a more accurate description of our measured parameter would be *“apparent permeability to calcein”* and we renamed permeability to this throughout the text.

2. The authors have done a creditable effort to demonstrate that Cx31 appears not to be anywhere near as effective in rescuing metabolite deficient phenotypes as Cx26. They also make the same claim for Cx43, but this is not supported by the data they present. They test if ALDO KO DLD1 (Figure 6) or NDUFS1-KO SW1222 (Figure 7) can be rescued by wt cells expressing Cx43 (C10), stating that "These co-cultures should, in theory, produce heterotypic connexins (should read gap junctions)". The problem is they should and likely do NOT, as Cx26 and 43 have been demonstrated to NOT form heterotypic channels, and the authors demonstrate earlier that neither DLD1 or SW1222 express Cx43. To demonstrate that Cx43 can or cannot mediate the metabolic rescue they are studying, they would need to do the Aldolase of NDUFS1 KOs in C10 cells, and then do cultures of the KO and wt cells, all of which express Cx43. Otherwise they need to limit their conclusions about lack of metabolic rescue to Cx31, which does form heteroptypic channels with Cx26. This is important, as Cx43 is also commonly expressed in tumors.

Thank you for this comment. We had, indeed, performed the Cx43 experiment. Figure 6H shows data from C10 cells (i.e. Cx43-dependent). However, we now noticed that the revised legend to Figure 6H had an error: the ALDOA KD was performed on C10 cells, *not DLD1 cells*. Nonetheless, the description of this experiment in the main text and in the figure labelling in Figure 6H are correct. The ALDOA KD of C10 is confirmed in the supplement. A similar ‘typo’ occurred for Figure 6I: the KD was in NCIH747 cells, *not DLD1 cells*. We profusely apologise for this oversight.

To recap: in the experiment shown in Figure 6H, we co-cultured wild-type cells with ALDOA-deficient C10 cells. The principal mechanism by which these monolayers are connected is via Cx43. However, these channels did not give rise to significant metabolic rescue, to the extent that Cx26-coupling does. In the text, we speculate that this may relate to our observation that Cx43 levels across a population of CRC cells is very heterogenous, and that only the positive sub-population would benefit from coupling. This, therefore, limits the extent of metabolic rescue across a cell-population (e.g. monolayer).

Thank you for drawing our attention to the inability of Cx43 and Cx26 channels to form heterotypic gap junctions. We have revised the text on p17 (line 390) to correct our mistake. We quote a review article.